# One-Step Gradient Delay is Not a Barrier for Large-Scale Asynchronous Pipeline Parallel LLM Pretraining

**Philip Zmushko** [1 2 3 * †]   **Egor Petrov** [2 3 *]   **Nursultan Abdullaev** [2 3 4]   **Mikhail Khrushchev** [2]   **Samuel Horváth** [5]

[*]Equal contribution. [†]Work completed while Philip Zmushko was at Yandex and BRAIn Lab; current affiliation: ISTA.

## Abstract

Modern large-scale LLM pretraining benefits from utilizing Pipeline Parallelism; however, synchronous implementations leave GPUs idle during pipeline bubbles, wasting computational resources. Asynchronous Pipeline Parallelism eliminates these bubbles, maximizing throughput at the cost of gradient staleness. Among asynchronous schedules, PipeDream-`2BW` is particularly appealing: unlike the original PipeDream schedule, it ensures a constant one-step gradient delay regardless of pipeline depth. However, its adoption remains limited due to the common belief that optimizing under staleness is fundamentally unstable. In this work, we challenge this assumption, demonstrating that degradation under one-step delay depends strongly on optimizer choice rather than being an intrinsic limitation. We provide the first comprehensive empirical analysis showing that while AdamW, the predominant optimizer at the time when PipeDream-`2BW` was introduced, indeed suffers from severe degradation, recent methods like Muon exhibit strong robustness under a one-step delay. We introduce an optimizer-agnostic Error Feedback-inspired correction to further mitigate delay effects. We provide supporting theoretical analysis demonstrating convergence for Muon with and without this correction. Extensive evaluation on models up to 10B parameters confirms that our strategy bridge the performance gap with synchronous training, highlighting the practical potential of asynchronous pipeline parallelism at scale.

[1]Institute of Science and Technology Austria (ISTA), Austria [2]Yandex, Russia [3]Basic Research of Artificial Intelligence Laboratory (BRAIn Lab), Russia [4]Innopolis University, Russia [5]Mohamed bin Zayed University of Artificial Intelligence (MBZUAI), UAE. Correspondence to: <zmushko.ph.a@gmail.com>.

*Proceedings of the $43^{rd}$ International Conference on Machine Learning*, Seoul, South Korea. PMLR 306, 2026. Copyright 2026 by the author(s).

## 1. Introduction

In the modern era of Large Language Models (LLMs), training on a single GPU is no longer feasible due to memory constraints, necessitating distributed training with model parallelism. One common approach is Pipeline Parallelism (PP) (Huang et al., 2019), which partitions the model vertically into stages. While PP was historically an essential component of large-scale training (Narayanan et al., 2021b), it became less popular following the introduction of memory-efficient data-parallel approaches like ZeRO (Rajbhandari et al., 2020), and was primarily used only for models larger than 70B parameters (Grattafiori et al., 2024). However, the rise of Mixture-of-Experts (MoE) architectures (Shazeer et al., 2017) has made this strategy less effective: MoE layers substantially increase the communication involved in training, without a proportional increase in per-layer computation. This lower compute-to-communication ratio has led to renewed interest in PP, in recent large-scale runs (Liu et al., 2024a;b; KimiTeam, 2025).

However, synchronous PP suffers from a fundamental limitation: preserving synchronous parameter updates introduces empty slots in the pipeline schedule, known as "bubbles", during which some GPUs remain idle, reducing global utilization and efficiency. Despite extensive efforts to mitigate these bubbles (Narayanan et al., 2021b; Qi et al., 2023; Liu et al., 2024c), they cannot be entirely eliminated in a synchronous setting. Alternatively, Asynchronous PP (Async PP) avoids synchronization entirely, allowing for the complete removal of pipeline bubbles. Under standard bubble models, this can translate into substantial schedule-level speedups over synchronous PP[1]. Unfortunately, this comes at a cost: Async PP is no longer semantically equivalent to conventional minibatch training, since gradients may be computed using stale parameters or applied after a delay.

Unlike synchronous PP, Async PP remains significantly less explored in the context of language model pre-training. To the best of our knowledge, Ajanthan et al. (2025) is the only work that studies Async PP for pre-training decoder-only language models. However, their experiments rely on

---

[1]See Section E.2 for a quantitative estimate.

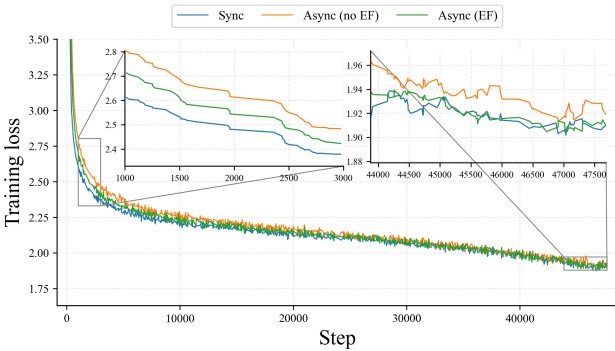

**Figure 1.** Training loss on a 10B MoE model trained for 200B FineWeb-Edu tokens. Both asynchronous runs remain stable throughout training, with Async PP + Error Feedback closing the final gap to the synchronous baseline entirely.

the original PipeDream schedule (Narayanan et al., 2019), which has a critical limitation: **variable gradient delays**. Because PipeDream updates parameters immediately after each local backward pass, different pipeline stages observe gradients with different amounts of delay. This staleness heterogeneity leads to severe convergence degradation as the number of stages increases: Ajanthan et al. (2025) report an increase of more than $0.2$ in validation loss compared to synchronous training at $16$ stages, a practical scale for real-world training (Liu et al., 2024b).

These findings highlight the need for an asynchronous approach to language modeling that remains robust as the pipeline depth increases. PipeDream-2BW (Narayanan et al., 2021a) is a natural candidate, as it ensures a constant gradient delay across all stages. By performing updates once every $M$ backward passes, PipeDream-2BW guarantees a **uniform staleness of 1 regardless of the pipeline size**. Rather than dealing with variable delays across stages, this reduces the optimization challenge to the cleaner setting of training with a fixed one-step delay: $w_{t+1} = w_t - u_{t-1}(g_{t-1})$, where $u_{t-1}$ denotes the optimizer update function applied to the delayed gradient $g_{t-1}$[2] While optimization under staleness is well studied in theory (Mishchenko et al., 2022; Koloskova et al., 2022), its practical application to LLM pre-training remains an open challenge. We address this gap by providing practical guidance on the effects of gradient delay and demonstrating the practical viability of Async PP for LLM pre-training.

Our contributions can be summarized as follows:

- We conduct the first comprehensive empirical analysis of optimizers and hyperparameters for language model training under gradient staleness, identifying a critical relationship between momentum and loss degradation. In particular, we show that AdamW (Loshchilov & Hutter,

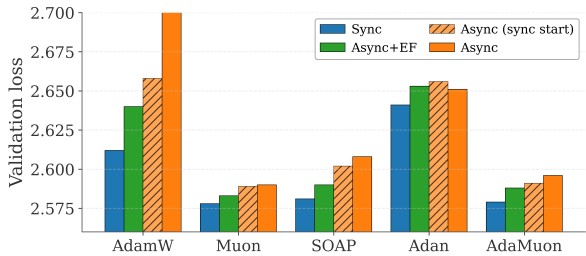

**Figure 2.** Validation loss of synchronous and one-step delayed optimizers on the 360M model. For most optimizers, Error Feedback cuts the sync-async gap by more than half compared to standard delayed training and outperforms the synchronous-start baseline.

2017), the historically dominant optimizer during the development of early Async PP methods, suffers substantial quality loss under staleness. In contrast, several modern optimizers remain surprisingly robust. Among them, Muon (Jordan et al., 2024), which is rapidly emerging as a leading optimizer for LLM pre-training (KimiTeam, 2025; Zeng et al., 2025), offers a particularly strong trade-off: it achieves competitive synchronous performance while maintaining a small sync-async gap under default hyperparameters.

- We investigate several staleness mitigation strategies and propose an optimizer-agnostic correction mechanism inspired by Error Feedback (Seide et al., 2014). This technique consistently narrows the performance gap between synchronous and asynchronous and further improves the already small gap observed for Muon.

- We empirically demonstrate the superiority of constant gradient delay over variable delay by comparing PipeDream-2BW against the original PipeDream schedule across multiple optimizers. These results show that fixed staleness is crucial for stable Async PP training at larger pipeline depths.

- We provide theoretical convergence guarantees for Muon with our Error-Feedback correction under gradient staleness. To the best of our knowledge, this constitutes the first convergence analysis of LMO algorithms with Error Feedback under fixed gradient delay.[3]

- Finally, we validate our findings at scale by training a 10B-parameter Mixture-of-Experts (MoE) model on 200B tokens with Muon. **With Async PP and Error Feedback, we achieve a final loss identical to that of the synchronous baseline while using the exact same hyperparameters**, marking, to the best of our knowledge, the first successful demonstration of Async PP at this scale without quality degradation.

---

[2]The time index $t$ in $u_t$ indicates that the update may depend on iteration-dependent quantities, such as the learning rate, momentum, or variance buffers.

[3]Concurrent work Sadiev et al. (2026) studies asynchronous LMO methods with delay thresholding in heterogeneous server-worker settings; see Section 7 for a discussion.

## 2. One-Step Delayed Optimization

Before evaluating optimizers, we first clarify the delayed-update abstraction used throughout the paper. The closest prior work on Async PP for language model pre-training, Ajanthan et al. (2025), relies on the original PipeDream schedule (Narayanan et al., 2019), thereby inheriting its variable gradient delays. In PipeDream, each stage updates its parameters immediately after a local backward pass, so different stages can observe gradients with different amounts of delay. Moreover, preserving forward-backward consistency requires weight stashing, with one stashed parameter version per delay level induced by the schedule. We instead use PipeDream-2BW (Narayanan et al., 2021a), which avoids variable delays by updating stage parameters only after a full minibatch of $M$ micro-batches has completed backward propagation. Intuitively, when $M \geq P-1$, where $P$ is the number of pipeline stages, each micro-batch has enough time to complete its forward-backward pass through the pipeline before the next minibatch triggers an update, yielding a uniform one-step gradient delay across all stages. PipeDream-2BW also reduces weight stashing to a single additional parameter copy, whose memory cost is negligible in realistic LLM training setups; see Section E.1 for a quantitative discussion. As shown by Narayanan et al. (2021a) for SGD, this constant-delay schedule can be viewed as standard optimization with the previous-step gradient. Extending this view from SGD to an arbitrary optimizer gives the generic delayed-update rule in Algorithm 1.

---

**Algorithm 1** Delayed Gradient Update

---

**Require:** Initial point $x_0$, learning rate $\eta$, iterations $T$
**Ensure:** Final point $x_T$
 1: Initialize $g_0 = g_{-1} = 0$
 2: **for** $t = 0, 1, \ldots, T-1$ **do**
 3:     Compute gradient $g_t$
 4:     Update optimizer statistics with $g_{t-1}$
 5:     Calculate update step $u_{t-1}(g_{t-1})$
 6:     **if** Standard Async **then**
 7:         $x_{t+1} \leftarrow x_t - u_{t-1}(g_{t-1})$
 8:     **else if** Error-Feedback (Section 3.2) **then**
 9:         $x_{t+1} \leftarrow x_t - 2 \cdot u_{t-1}(g_{t-1}) + u_{t-2}(g_{t-2})$
10:     **end if**
11: **end for**

---

We begin our empirical analysis by evaluating existing language-model optimizers under this one-step delayed-update rule. Full details on models, data, training protocol and hyperparameter grids are provided in Appendix C.

### 2.1. Initial Observation: The Staleness Robustness Gap

We start with a simple comparison between two representative optimizers: AdamW (Kingma, 2014; Loshchilov & Hutter, 2017) and Muon (Jordan et al., 2024; Liu et al.,

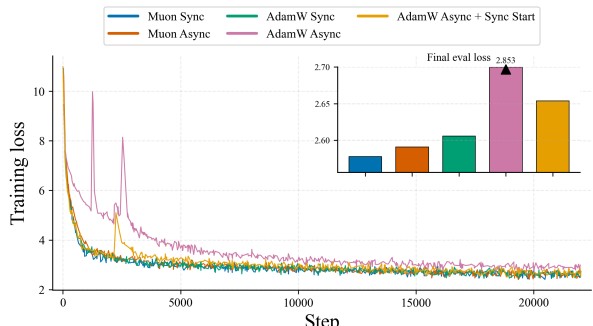

**Figure 3.** Synchronous and delayed AdamW and Muon training on the 360M model. AdamW degrades substantially under one-step delay, even with a synchronous start, whereas Muon achieves a much smaller final sync-async gap.

2025a). AdamW is the standard optimizer used in most LLM pre-training pipelines and was the dominant choice when early asynchronous pipeline-parallel methods were developed. Muon, in contrast, is a more recent optimizer that has seen rapid adoption in large-scale LLM training reports (KimiTeam, 2025; Zeng et al., 2025).

For this initial comparison, we train a 360M-parameter model and use commonly adopted default hyperparameters for both optimizers, tuning only the learning rate. Specifically, we use global weight decay $0.1$ and gradient clipping at $1.0$ for both methods. For AdamW, we use $\beta = (0.9, 0.95)$ (Touvron et al., 2023a;b; Li et al., 2025a; Liu et al., 2024b). For Muon, we use momentum $\mu = 0.95$ and update RMS $0.2$ (Liu et al., 2025a; Zeng et al., 2025).

The results in Figure 3 show a clear gap in robustness to one-step gradient delay. Both optimizers train competitively in the synchronous setting, but their delayed variants behave very differently. AdamW suffers severe quality degradation under one-step delay ($> 0.2$), with the training loss diverging early from the synchronous trajectory. Starting training synchronously before switching to delayed updates, a stabilization heuristic used in prior work (Ren et al., 2021), improves AdamW but does not close the gap: the final sync-async loss gap remains $0.046$, and the loss exhibits a sharp spike immediately after the switch to delayed training, as we analyze in more detail in Section 3.1. In contrast, Muon achieves a much smaller final gap of only $0.012$ without any synchronous start.

This initial observation suggests that staleness is not uniformly harmful across optimizers. Rather, robustness to one-step delay appears to depend strongly on the optimizer dynamics.

### 2.2. Hyperparameter Sensitivity and Benchmarking

We next broaden the comparison to a larger set of optimizers and study which hyperparameters control robustness to one-step delay. We evaluate AdamW (Loshchilov & Hut-

**Table 1.** Validation loss under synchronous and one-step delayed training for 135M and 360M models. Most modern optimizers remain robust to one-step delay, while AdamW and MARS suffer severe degradation. Colors indicate the loss increase relative to the synchronous baseline: green $\Delta \leq 0.015$, cyan $0.015 < \Delta \leq 0.03$, blue $0.03 < \Delta \leq 0.05$, and red $\Delta > 0.05$.

| | SmoLLM-135M | | SmoLLM-360M | |
| --- | --- | --- | --- | --- |
| **Optimizer** | **Sync** | **Async** | **Sync** | **Async** |
| Muon | 2.841 | 2.855 (+0.014) | 2.578 | 2.590 (+0.012) |
| Adan | 2.896 | 2.902 (+0.006) | 2.641 | 2.651 (+0.010) |
| AdaMuon | 2.845 | 2.867 (+0.022) | 2.579 | 2.596 (+0.017) |
| SOAP | 2.850 | 2.872 (+0.022) | 2.581 | 2.608 (+0.027) |
| Lion | 2.870 | 2.894 (+0.024) | 2.624 | 2.654 (+0.030) |
| MARS-M | 2.840 | 2.875 (+0.035) | 2.578 | 2.607 (+0.029) |
| NorMuon | 2.841 | 2.887 (+0.046) | 2.573 | 2.609 (+0.036) |
| NAdam | 2.896 | 2.936 (+0.040) | 2.651 | 2.694 (+0.043) |
| MARS | 2.874 | 3.343 (+0.469) | 2.615 | 2.897 (+0.282) |
| AdamW | 2.877 | 3.227 (+0.350) | 2.612 | 2.890 (+0.278) |

ter, 2017), Muon (Liu et al., 2025a), SOAP (Vyas et al., 2024), Nadam (Dozat, 2016), MARS (Yuan et al., 2024), Adan (Xie et al., 2024), and Lion (Chen et al., 2023), along-side several Muon variants including AdaMuon (Si et al., 2025), NorMuon (Li et al., 2025c), and MARS-M (Liu et al., 2025b). For AdamW, we first perform a broad grid search over learning rate and weight decay to establish a strong baseline. For the remaining optimizers, except Lion which operates on a different learning-rate scale, we search around the best AdamW settings, following prior evidence that many modern LLM optimizers have optima in similar regions (Semenov et al., 2025; Wen et al., 2025). We then varied both global and optimizer-specific hyperparameters on a 135M parameter model.

**The Role of Momentum.** Across the hyperparameters we tested, most showed no clear systematic effect on the sync-async performance gap. In contrast, one parameter exhibited a consistent trend across optimizers: the *momentum decay ratio*, denoted as $\mu$ in Muon-like optimizers and $\beta_1$ in Adam-like optimizers. Formally, this corresponds to the coefficient $\beta$ in the Exponential Moving Average (EMA) update: $m_t = \beta m_{t-1} + (1 - \beta)g_t$. As shown in Figure 4, increasing this coefficient consistently reduces the loss penalty caused by one-step delay[4].

This observation suggests a broader explanation for the benefits reported by Ajanthan et al. (2025). They observed that higher momentum improves delayed training and attributed this effect to the "look-ahead" structure of Nesterov Accelerated Gradient, motivating the use of Nadam (Dozat, 2016). Our ablations indicate that the effect is not specific to Nesterov-style updates: higher momentum also improves robustness for optimizers without a look-ahead mechanism. We hypothesize that the mechanism is more fundamental:

---

[4]Note that we exclude certain high-momentum configurations, such as $\beta_1 = 0.99$ for specific Adam variants, where the synchronous baseline itself diverges due to instability.

in the presence of delayed gradients, the optimizer cannot rely as heavily on the instantaneous update as in the synchronous setting. A higher momentum coefficient effectively dampens the noise introduced by staleness, forcing the optimization trajectory to rely more on the accumulated history rather than the potentially erratic current step.

**Benchmarking Results.** After the hyperparameter sweep, we select for each optimizer the configuration with the best delayed-training performance, subject to the constraint that its synchronous baseline remains within 0.01 of the global synchronous optimum. The results for 135M and 360M models are summarized in Table 1. An important outcome is that severe degradation is concentrated in only a small sub-set of optimizers: AdamW and MARS degrade substantially under one-step delay, while most other modern optimizers are far more robust. In particular, Muon, Adan, AdaMuon, SOAP, and Lion keep the final loss gap within 0.03 across both model sizes. Among these methods, Adan achieves the smallest sync-async gap, which is also consistent with the role of momentum: its best configuration uses the default high first-moment coefficient $\beta_1 = 0.98$, substantially larger than the standard choice of $\beta_1 = 0.9$ in Adam-like optimizers. Muon, however, offers the strongest practical trade-off: it is among the best synchronous optimizers, remains highly robust under delay, and is increasingly relevant for modern LLM pre-training.

Motivated by the unusually poor behavior of AdamW-like optimizers, we further investigate possible sources of their degradation in Appendix D.1. Although these experiments do not fully explain AdamW's degradation, one targeted ablation provides additional evidence for the role of first-moment dynamics: delaying only the first-moment update $m_t$ closely reproduces the behavior of fully delayed AdamW.

Overall, the results in this section show that one-step staleness affects optimizers very differently, with momentum playing a particularly important role. They also highlight Muon as a particularly strong candidate for large-scale Async PP: it combines strong synchronous performance, robustness to delayed updates, and relevance to modern LLM pre-training. Before moving to scale, we next ask whether the remaining sync-async gap can be reduced by optimizer-agnostic mitigation strategies.

## 3. Staleness Mitigation

We now study generic mitigation strategies that can be applied on top of the delayed-update rule in Algorithm 1. We first evaluate several natural baseline strategies for mitigating one-step staleness, and then introduce an Error-Feedback-inspired correction that provides a more consistent improvement.

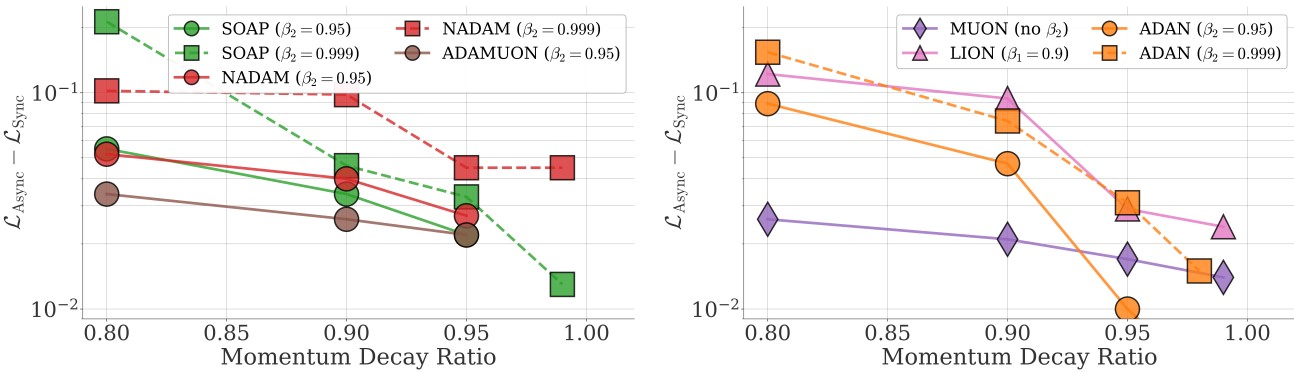

**Figure 4.** Final loss gap between synchronous and asynchronous training as a function of the momentum decay for various optimizers.

### 3.1. Baseline Mitigation Strategies

**Synchronous Start.** We first revisit the synchronous-start baseline discussed in Section 2.1. This strategy follows ZeRO-Offload (Ren et al., 2021), where starting training synchronously before switching to asynchronous updates was reported to improve stability. Here, we evaluate whether the same strategy provides a consistent mitigation across the broader optimizer set.

The results in Table 2 show a mixed picture. For many optimizers, synchronous start recovers a nontrivial fraction of the remaining sync-async gap, typically around 20–30%. At the same time, as already noted in Section 2.1, this strategy introduces an additional sensitivity at the transition point. In particular, for adaptive optimizers such as SOAP and Adan, larger $\beta_2$ values can make the switch from synchronous to delayed updates highly unstable. At the switching point, these runs exhibit a sharp loss spike, followed by a long recovery period and, in some cases, divergence (see example in Figure 11). Beyond these stability issues, synchronous start temporarily reintroduces pipeline bubbles and requires supporting both synchronous and asynchronous execution modes, reducing the practical throughput benefit of Async PP. Overall, we find synchronous start to be a useful baseline, but not a reliable standalone mitigation strategy.

**Synchronous Cooldown.** We also evaluate the opposite schedule-level intervention: switching from delayed updates back to synchronous training near the end of the run. This tests whether removing staleness in the final phase can recover the remaining loss gap. As shown in Table 7, this strategy yields only marginal improvements for both Muon and AdamW. Thus, the residual gap is not easily removed by making only the final part of training synchronous.

**DC-ASGD / Taylor-based Delay Compensation.** Finally, we test Delay-Compensated ASGD (DC-ASGD) (Zheng et al., 2017), which modifies the stale gradient using a Taylor-style correction term proportional to $\lambda \odot g^2 \odot \Delta w$. A simple scale estimate suggests that this correction is ex-

tremely small for LLM training unless $\lambda$ is very large: in our runs, gradients are typically around $10^{-5}$, while parameter updates are proportional to the learning rate, around $10^{-3}$. We therefore sweep $\lambda$ from $10^4$ to $10^8$. As shown in Figure 8, values up to $10^6$ produce losses indistinguishable from the standard delayed baseline up to the third decimal place, indicating that the correction remains too small to matter. Larger values make the correction visible, but only degrade training rather than improving it. We therefore find this gradient-level correction ineffective in our setting.

Taken together, these baselines suggest that simple schedule-level or gradient-level fixes are insufficient. Synchronous start can help some optimizers but may introduce a sharp switching spike. Synchronous cooldown has little effect, while Taylor-based correction only destabilizes training once scaled up. We therefore turn to a correction mechanism that operates directly at the optimizer-update level.

### 3.2. Error Feedback

To further reduce the effect of stale gradients, we introduce a lightweight correction inspired by Error Feedback (Seide et al., 2014; Stich & Karimireddy, 2019). At step $t$, standard delayed training applies the update $-u_{t-1}(g_{t-1})$, even though the update that would have matched the current gradient is not available until one step later. Once the next gradient is computed, we can retrospectively estimate the update error caused by staleness. In particular, at step $t$, the previous update *actually applied* was based on $g_{t-2}$, whereas the update we *would have applied* with fresh information would be based on $g_{t-1}$. The difference between these two parameter updates is therefore $u_{t-2}(g_{t-2}) - u_{t-1}(g_{t-1})$. Adding this correction to the current delayed update gives

$$x_{t+1} = x_t - \underbrace{u_{t-1}(g_{t-1})}_{\text{Async Update}} + \underbrace{(u_{t-2}(g_{t-2}) - u_{t-1}(g_{t-1}))}_{\text{Error Correction}}$$

$$= x_t - 2 \cdot u_{t-1}(g_{t-1}) + u_{t-2}(g_{t-2}) \qquad (1)$$

**Table 2.** Validation loss under synchronous and delayed training with different staleness mitigation techniques for 360M model. The **Standard** column is delayed training without mitigation; percentages in parentheses show the recovered fraction of the Standard sync-async gap. Error Feedback recovers more than half of the gap for most optimizers.

| Optimizer | Sync | Async | | |
|---|---|---|---|---|
| | | EF | Sync Start | Standard |
| Muon | 2.578 | **2.583** (-71%) | 2.589 (-8%) | 2.590 |
| Adan | 2.641 | 2.653 (+20%) | 2.656 (+50%) | **2.651** |
| AdaMuon | 2.579 | **2.588** (-53%) | 2.591 (-29%) | 2.596 |
| SOAP | 2.581 | **2.590** (-67%) | 2.602 (-22%) | 2.608 |
| Lion | 2.624 | 2.642 (-40%) | **2.639** (-50%) | 2.654 |
| NorMuon | 2.573 | **2.584** (-69%) | 2.596 (-36%) | 2.609 |
| NAdam | 2.651 | 2.703 (+21%) | 2.695 (+2%) | **2.694** |
| MARS | 2.615 | **2.657** (-85%) | 2.820 (-27%) | 2.897 |
| AdamW | 2.612 | **2.640** (-90%) | 2.658 (-84%) | 2.890 |

Despite the simplicity of this formulation, to the best of our knowledge, this specific update-level correction scheme has not been previously proposed in the literature.

Table 2 and Figure 2 show that this correction provides a more consistent benefit than the baseline strategies above. For several robust optimizers, including Muon, AdaMuon, SOAP, and NorMuon, Error Feedback recovers roughly 50–70% of the degradation introduced by delayed training. It also substantially improves the most degraded AdamW-like runs, recovering 85–90% of the gap for MARS and AdamW in our 360M experiments. The method is not universally beneficial: it slightly degrades Adan and NAdam in this benchmark. Nevertheless, across the full optimizer set, it is the most reliable mitigation strategy we evaluate.

Two practical details are worth noting. The proposed Error Feedback approach stores one additional model-sized buffer, but this adds only a small constant memory overhead in realistic LLM training setups; see Appendix E.1. A similar correction can also be applied to raw gradients before passing them to the optimizer, but this variant diverges in our experiments; see Figure 9.

Taken together, Error Feedback provides a lightweight and broadly applicable way to reduce this gap. Combined with the optimizer study in Section 2, these results make Muon a natural choice for our large-scale validation: it is competitive in the synchronous setting, robust under one-step delay, and benefits from the proposed correction. Before moving to the large-scale experiment, we provide a theoretical analysis of Muon under delayed updates.

## 4. Theoretical analysis

The empirical results above suggest that Muon is a promising optimizer for one-step delayed training. We analyze Muon-style Linear Minimization Oracle (LMO) updates under gradient staleness. While convergence under delayed gradients has been studied for other algorithm fami-

---

**Algorithm 2** Delayed Muon

1: **input:** $\mathbf{X}_0, \mathbf{M}_0 \in \mathbb{R}^{m \times n}$
2: **parameters:** stepsize $\eta > 0$, momentum $\mu \in (0,1)$, weight decay $\lambda \in (0,1)$, number of iterations $T$
3: **for** $t = 0, 1, \ldots, T-1$ **do**
4:     Compute gradient: $\mathbf{G}_{t-1} \leftarrow \nabla f(\mathbf{X}_{t-1}, \xi_{t-1})$
5:     $\mathbf{M}_{t-1} \leftarrow (1-\mu)\mathbf{M}_{t-2} + \mu\mathbf{G}_{t-1}$
6:     $\mathbf{O}_{t-1} \leftarrow \text{Newton-Schulz}(\mathbf{M}_{t-1})$
7:     $\mathbf{U}_{t-1} \leftarrow \eta(\mathbf{O}_{t-1} + \lambda\mathbf{X}_t)$
8:     **if** Standard Async **then**
9:         $\mathbf{X}_{t+1} \leftarrow \mathbf{X}_t - \mathbf{U}_{t-1}$
10:     **else if** Error-Feedback (Section 3.2) **then**
11:         $\mathbf{X}_{t+1} \leftarrow \mathbf{X}_t - 2\mathbf{U}_{t-1} + \mathbf{U}_{t-2}$
12:     **end if**
13: **end for**
14: **output:** $\mathbf{X}_T$

---

lies (Mishchenko et al., 2022; Koloskova et al., 2022), theoretical guarantees for LMO-based methods under the fixed-delay updates considered here remain limited. To the best of our knowledge, this is the first convergence analysis of Muon with Error Feedback under fixed gradient delay.[5] In the main text, we focus on Muon as the most relevant instance for our experiments, while Appendix A gives the general formulation for arbitrary norms and the full convergence proofs for arbitrary delay $\tau \geq 0$.

First, we theoretically formulate the optimization problem for Muon, following the setting covered in (Kovalev, 2025):

$$\min_{\mathbf{X} \in \mathbb{R}^{m \times n}} f(\mathbf{X}) \tag{2}$$

where $f(\cdot) \colon \mathbb{R}^{m \times n} \to \mathbb{R}$ is a bounded from below and differentiable objective function.

**Assumption 4.1.** For further theoretical analysis, we consider the following:

1. **Stochastic gradient estimator.**
   $\mathbb{E}_\xi[\nabla f(\mathbf{X}; \xi)] = \nabla f(\mathbf{X})$,
   $\mathbb{E}_\xi[\|\nabla f(\mathbf{X}; \xi) - \nabla f(\mathbf{X})\|_2^2] \leq \sigma^2$.
2. **Smoothness.** $\|\nabla f(\mathbf{X}) - \nabla f(\mathbf{X}')\|_{\text{nuc}} \leq L\|\mathbf{X} - \mathbf{X}'\|_{op}$.
3. **Star convexity.**
   $f(\alpha\mathbf{X}^* + (1-\alpha)\mathbf{X}) \leq \alpha f(\mathbf{X}^*) + (1-\alpha)f(\mathbf{X})$.

for any $\mathbf{X}, \mathbf{X}' \in \mathbb{R}^{m,n}$, where $\mathbf{X}^* \in \mathbb{R}^{m \times n}, \alpha \in (0,1)$ and $\sigma \geq 0$.

These assumptions have been widely adopted for the analysis of many stochastic gradient optimization algorithms (Gower et al., 2019; Horváth et al., 2023; Kovalev, 2025)

Then, we introduce Muon version with gradient delay, formulated in Algorithm 2.

---

[5] Concurrent work Sadiev et al. (2026) studies asynchronous LMO methods with delay thresholding in heterogeneous server-worker settings; see Section 7 for a discussion.

**Theorem 4.2** (Delayed Muon with Weight Decay). *Let Assumption 4.1 hold, and let $\mathbf{M}_0 = \mathbf{G}(\mathbf{X}_0)$. Then the iterations of Algorithm 2 with Weight Decay $\lambda > 0$ satisfy:*

$$\mathbb{E}[f(\mathbf{X}_T) - f(\mathbf{X}^*)] \leq (1-\lambda)^K (f(\mathbf{X}_0) - f(\mathbf{X}^*))$$
$$+ 2\eta \left( \frac{\rho\sigma}{\mu} + \frac{\sqrt{2\mu}\sqrt{\rho^2\sigma^2 + 8(L\eta)^2}}{\lambda} \right) + \frac{4L\eta^2}{\lambda}\left(1 + \frac{1}{\mu}\right).$$

*where $\rho = \sqrt{\min(m,n)}$, and $\eta$, $\lambda$ satisfy the following:*

$$\eta \geq \lambda \max\{\|\mathbf{X}_0\|, \|\mathbf{X}^*\|\} \tag{3}$$

*Proof.* This result is a direct corollary of our general convergence guarantee for delayed LMO algorithms, presented in Theorem A.6. The proof follows by instantiating the general theorem with the specific choices for the Muon optimizer: setting the regularizer $R(\mathbf{X}) \equiv 0$, using the operator norm $\|\cdot\|_{\mathrm{op}}$ and its dual, the nuclear norm $\|\cdot\|_{\mathrm{nuc}}$, and substituting the concrete norm equivalence constant $\rho = \sqrt{\min(m,n)}$. For a complete derivation of the general case, we refer the reader to Appendix A. $\square$

**Discussion.** The main difference between the obtained estimation for the delayed setup and the synchronous one lies in the noise bound, specifically $\sqrt{2\mu}\sqrt{\rho^2\sigma^2 + 8(L\eta)^2}$ for the delayed setup versus $\sqrt{\mu}\rho\sigma$ for the standard one. Following Corollary 2 from (Kovalev, 2025), where $\eta = \mathcal{O}\left(\min\{\frac{\epsilon}{L}, \frac{\epsilon^2}{\rho^2\sigma^2 L}\}\right)$ and $\mu = \mathcal{O}\left(\min\{1, \frac{\epsilon^2}{\rho^2\sigma^2}\}\right)$, one can show that the additional term caused by delayed gradients is generally small.

## 5. Large Scale Experiments

Having studied optimizer robustness, mitigation strategies, and theoretical guarantees under one-step delay, we now ask whether these findings transfer to realistic pre-training runs. This question is particularly important because the throughput benefits of Async PP are most relevant in large distributed training regimes, where pipeline bubbles translate into substantial wasted accelerator time (see Section E.2). Since benchmarking all optimizers at this scale is prohibitively expensive, we focus on **Muon**: across the previous sections, it combines strong synchronous performance, robustness to one-step delay, compatibility with Error Feedback, and convergence guarantees under staleness. We scale our evaluation to 2B- and 10B-parameter MoE (Shazeer et al., 2017) models to test whether Async PP can match synchronous training quality in realistic training scenarios.

### 5.1. 2B MoE Experiments

We train a MoE model with 2B total parameters and 500M active parameters for training horizons ranging from 50B to 200B tokens. To keep the global batch size close to

**Table 3.** Large-scale pretraining results: final validation loss for 10B MoE model trained for 200B tokens on the Fine-Web dataset.

| Optimizer | Sync | Async | Async + EF |
|-----------|------|-------|------------|
| Muon | **1.906** | 1.911 | **1.906** |

the optimum as the training horizon increases, we scale it according to $B \propto D^{0.58}$ following Li et al. (2025b), using $B = 1\text{M}$ tokens at $D = 50\text{B}$ as the anchor point.

**Learning Rate Robustness.** We first test whether the async performance gap is sensitive to changes in the learning rate. For each training horizon $D$, we sweep the peak learning rate and report the full results in Section D.7. Across this sweep, the sync-async gap remains stable: changing the learning rate shifts both synchronous and asynchronous losses similarly, without any sign of delayed-training-specific sensitivity.

**Scaling with Training Horizon.** We next test whether the effect of staleness grows as training progresses to longer horizons and lower losses. A natural concern is that delayed gradients may become increasingly harmful as training approaches convergence, where the optimization trajectory may require more accurate gradient information to continue reducing the loss. Using the learning-rate sweeps described above, we take the best validation loss at each training horizon and observe nearly parallel synchronous and asynchronous scaling curves in Figure 5. This indicates that one-step staleness does not introduce a growing optimization barrier between 50B and 200B tokens. Error Feedback also consistently recovers a substantial fraction of the remaining gap across all tested scales. While verifying this behavior at trillion-token scale remains an important direction for future work, the absence of gap growth up to 200B tokens supports the scalability of Async PP in realistic pre-training runs.

### 5.2. 10B MoE Experiments

To test whether our findings hold at the largest scale available in our experiments, we train a 10B-parameter Mixture-of-Experts model. The model uses a Qwen3-Next-like architecture (QwenTeam, 2025) with Gated Delta Net layers (Yang et al., 2024). We train for 200B tokens with a global batch size of 4M tokens and a peak learning rate of 0.00225, comparing the synchronous baseline against standard Async PP and Async PP with Error Feedback.

The training loss trajectories are shown in Figure 1, and the final validation losses are reported in Table 3. Standard Async PP remains highly competitive at this scale, incurring only a small final loss gap relative to the synchronous baseline (1.911 vs. 1.906). **With Error Feedback, Async PP closes this gap entirely, matching the synchronous final loss** of 1.906 while using the exact same hyperparameters.

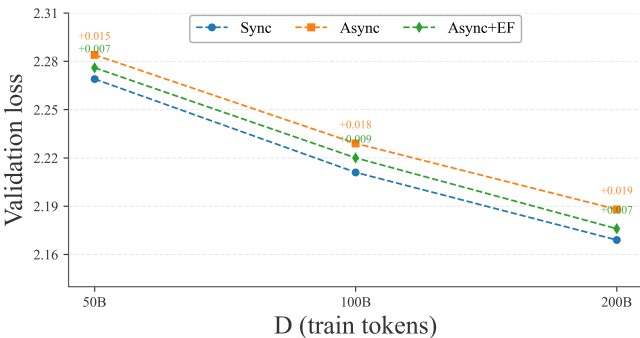

**Figure 5.** Validation loss of the 2B MoE model across training horizons. The synchronous and asynchronous scaling curves remain nearly parallel from 50B to 200B tokens, indicating no growth of the sync-async gap with longer training; Error Feedback consistently reduces the remaining gap.

Notably, the relative degradation at 10B scale is smaller than in our smaller dense-model experiments, suggesting that one-step delayed optimization remains robust in realistic large-scale MoE pre-training. Both asynchronous runs remain stable throughout training, despite a small lag during the early phase.

To the best of our knowledge, this is the first successful demonstration of Async PP on a model of this scale without quality degradation. Together with the throughput motivation of Async PP, these results highlight the practical potential of asynchronous pipeline parallelism for large-scale LLM pre-training.

## 6. Comparison with PipeDream

While the main experiments use PipeDream-2BW (Narayanan et al., 2021a), prior work on Async PP for language model pre-training (Ajanthan et al., 2025) used the original PipeDream schedule (Narayanan et al., 2019). In that setting, Ajanthan et al. (2025) found that Nadam can perform reasonably at small pipeline depths but degrades substantially as the number of stages increases. This leaves open whether the degradation is primarily due to the optimizer choice or to the PipeDream-style variable-delay schedule itself. In particular, if the instability is mostly optimizer-driven, the more robust optimizers identified in Section 2 could make the original PipeDream schedule a practical alternative. We therefore compare against the original PipeDream schedule using these optimizers and also evaluate whether EF can further reduce the degradation.

### 6.1. Experimental Results

We evaluate the original PipeDream schedule with $P \in \{4, 8, 16\}$ stages using Muon, SOAP, and Nadam with the best hyperparameter configurations from Section 2. To account for the schedule's mechanics, we set the effective

**Table 4.** Comparison of synchronous training, PipeDream-2BW with constant one-step delay, and the original PipeDream schedule with variable delay on the 135M model. Bold entries indicate the smallest loss gap relative to the corresponding synchronous baseline within each asynchronous column.

| Optimizer | Sync | 2BW | PipeDream | | |
|---|---|---|---|---|---|
| | | const | $P = 4$ | $P = 8$ | $P = 16$ |
| Nadam | 2.891 | 2.936 | **2.889** | 2.910 | 2.950 |
| Muon, 0.95 | 2.839 | 2.856 | 2.861 | 2.881 | 2.917 |
| Muon, 0.99 | 2.841 | 2.855 | 2.844 | **2.857** | **2.882** |
| SOAP | 2.855 | **2.868** | 2.864 | 2.881 | 2.910 |

batch size per update to $B_{\text{sync}}/P$. This smaller per-update batch arises because the original PipeDream schedule performs an optimizer step after every backward pass, whereas PipeDream-2BW accumulates gradients over a full minibatch before applying an update. Thus, unlike PipeDream-2BW, original PipeDream does not preserve the same effective global batch size per weight update as the synchronous baseline. This makes the comparison intentionally faithful to the original schedule, but also highlights a practical drawback: original PipeDream may require separate batch-size and hyperparameter calibration, whereas PipeDream-2BW can directly reuse choices derived from synchronous scaling laws for optimal and critical batch sizes (Zhang et al., 2024; Merrill et al., 2025).

The baseline results without EF are presented in Table 4, with the corresponding Error-Feedback ablations in Table 8. The results show that the main trends from Sections 2 and 3.2 carry over to the original PipeDream schedule. Muon and SOAP are generally more robust than Nadam, especially at larger pipeline depths. For Muon, increasing the momentum from $\mu = 0.95$ to $\mu = 0.99$ improves performance at every pipeline depth, both with and without EF, further supporting the role of momentum observed in Section 2.2. EF also provides small but consistent improvements for Muon and SOAP, while not consistently helping Nadam, in line with the pattern observed in Table 2. At shallow pipeline depth, these improvements can nearly close the gap: for $P = 4$, Muon with $\mu = 0.99$ and EF reaches 2.840, matching the synchronous baseline within noise (2.841), while SOAP reaches 2.858 compared to its synchronous baseline of 2.855.

However, these gains do not remove the scaling issue of the original PipeDream schedule. As the number of stages increases, all methods degrade substantially: at $P = 16$, even the best configuration, Muon with $\mu = 0.99$ and EF, loses more than 0.03 relative to its synchronous baseline. These results suggest that robust optimizers and EF can make the original PipeDream schedule viable for shallow pipelines, but are insufficient at larger pipeline depths. Taken together, these results reinforce the central role of PipeDream-2BW: robust optimizers can partially compensate for the original PipeDream schedule at small pipeline depths, but scalable

Async PP requires the constant-delay guarantees provided by PipeDream-`2BW`.

## 7. Related work

**Asynchronous Pipeline Parallelism.** The domain of Asynchronous Pipeline Parallelism was established by PipeDream (Narayanan et al., 2019), which utilized weight stashing to ensure consistent weights for forward and backward passes, albeit yielding variable gradient staleness. Subsequent approaches like PipeMare (Yang et al., 2021), SpecTrain (Chen et al., 2018), XPipe (Guan et al., 2019) and PipeOptim (Guan et al., 2025) prioritized memory efficiency by removing stashing; however, these methods fundamentally compromise optimization integrity by allowing forward and backward passes to execute on different model versions. In the context of language modeling, Ajanthan et al. (2025) were the first to demonstrate the viability of Async PP. Despite this milestone, their work relies on the original PipeDream schedule, thereby inheriting its critical drawback of *variable gradient delays across pipeline stages*. To address these issues, we adopt the PipeDream-2BW (Narayanan et al., 2021a) scheme. The fundamental distinction of this approach is that it accumulates gradients over $M$ micro-batches rather than performing parameter updates after every step. This design enables it to achieve a *constant delay of 1 across all pipeline stages*, at the cost of maintaining just a single additional copy of model parameters in memory.

**Optimizer Benchmarking.** Recently, the community has placed increased emphasis on the empirical evaluation of optimization algorithms for LLMs. Studies like Semenov et al. (2025) and Wen et al. (2025) provide extensive benchmarking regarding convergence and performance, while Vlassis et al. (2025) explores optimizer interactions with quantization. We complement this line of work by conducting a comprehensive benchmark of optimizers specifically under the constraints of asynchronous gradient delay.

**Error Feedback.** Error-Feedback (EF) was originally introduced by Seide et al. (2014) to compensate for quantization errors. Since then, it has been extensively utilized in the context of gradient compression (Stich et al., 2018; Alistarh et al., 2018; Karimireddy et al., 2019). Recently, Gruntkowska et al. (2025) investigated EF with the Muon optimizer under compression constraints, while Stich & Karimireddy (2019) considered the interplay between EF and gradient delays. Our work uniquely synthesizes these directions by applying EF specifically to address the staleness in gradients.

**Optimization with Delayed Gradients.** The theoretical foundations of optimization under gradient delays are well-established (Agarwal & Duchi, 2011; Mishchenko et al., 2022; Koloskova et al., 2022), with research in this domain

continuing to evolve (Maranjyan et al., 2025). Stale updates have also been studied in systems-motivated distributed optimization frameworks, such as Pipe-SGD (Li et al., 2018), which pipelines AllReduce-based data-parallel training and provides convergence guarantees for convex and strongly convex objectives. More closely related to our mitigation study, SAPipe (Chen et al., 2022) introduces staleness-aware compensation for data-parallel training, where staleness arises from delayed gradient aggregation across workers rather than from asynchronous pipeline-parallel model updates. Their approach relies on weight prediction, whereas our main mitigation mechanism is an update-level Error-Feedback correction. We include a small-scale comparison with a SAPipe-style weight-prediction correction in Table 6. This correction performs on par with our Error-Feedback mechanism, indicating that weight prediction is another effective mitigation strategy for one-step staleness. Since we became aware of SAPipe only after completing the main experimental study, we leave a more extensive evaluation of SAPipe-style corrections in Async PP to future work. We distinguish our work from this broader line by conducting an extensive empirical investigation of delayed optimization across modern LLM optimizers and validating Async PP at scale.

**Concurrent Work on Asynchronous LMO Methods.** Concurrent with our work, Sadiev et al. (2026) study asynchronous LMO optimization in heterogeneous server-worker systems. Their setting is complementary to ours: Ringmaster LMO handles variable delays caused by heterogeneous worker runtimes via delay thresholding, whereas we focus on the fixed-delay regime induced by PipeDream-`2BW` in asynchronous pipeline-parallel training and cover the Error-Feedback correction used in our experiments. Experimentally, their evaluation focuses on stochastic quadratic problems and simulated-worker NanoChat pre-training, whereas ours studies large-scale LLM pre-training under Async PP, including optimizer benchmarking, staleness mitigation, and validation up to 10B parameters.

## 8. Discussion

This work presents a comprehensive analysis of optimizer dynamics under gradient staleness, highlighting the sensitivity of standard algorithms like AdamW contrasted with the stability of modern alternatives such as Muon. By utilizing PipeDream-`2BW` alongside a proposed Error-Feedback correction, we establish a framework that effectively mitigates convergence degradation. Our theoretical analysis provides convergence guarantees for these methods, while *empirical validation on a 10B parameter Mixture-of-Experts model* demonstrates performance parity with synchronous baselines. These results indicate that with appropriate algorithmic choices, asynchronous pipeline parallelism offers a viable and efficient pathway for large-scale model training.

# Acknowledgements

This work was conducted while Philip Zmushko was affiliated with Yandex and BRAIn Lab; he is currently affiliated with ISTA. The work of Egor Petrov was supported by the Ministry of Economic Development of the Russian Federation (agreement No. 139-15-2025-013, dated June 20, 2025, IGK 000000C313925P4B0002).

We thank Aleksandr Beznosikov and our colleagues from Yandex Research, Yandex, and BRAIn Lab for fruitful discussions.

# Impact Statement

This paper presents work whose goal is to advance the field of Machine Learning. There are many potential societal consequences of our work, none which we feel must be specifically highlighted here.

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

## A. Delayed Stochastic Non-Euclidean Trust-Region Theory

First, we theoretically formulate the general optimization problem, following the setting covered in (Kovalev, 2025):

$$\min_{x \in \mathcal{X}}[F(x) = f(x) + R(x)] \tag{4}$$

where $\mathcal{X}$ is a finite-dimensional vector space endowed with the inner product $\langle \cdot, \cdot \rangle \colon \mathcal{X} \times \mathcal{X} \to \mathbb{R}$, $f(\cdot) \colon \mathcal{X} \to \mathbb{R}$ is a bounded from below and differentiable objective function, and $R(\cdot) \colon \mathcal{X} \to \mathbb{R} \cup \{+\infty\}$ is a proper convex regularizer. For further theoretical analysis, we consider the following assumptions

**Assumption A.1** (Stochastic gradient estimator). We assume access to an unbiased stochastic gradient estimator $g(x; \xi)$ with bounded variance, for which the following holds for all $x \in \mathcal{X}$:

$$\mathbb{E}_{\xi \sim \mathcal{D}} g(x; \xi) = \nabla f(x)$$
$$\mathbb{E}_{\xi \sim \mathcal{D}} \|g(x; \xi) - \nabla f(x)\|_2^2 \leq \sigma^2$$

**Assumption A.2** (Smoothness). We assume that function $f(\cdot)$ has Lipschitz gradient with respect to the considered vector space $\mathcal{X}$:

$$\|\nabla f(x) - \nabla f(x')\|_* \leq L\|x - x'\| \text{ for all } x, x' \in \mathcal{X}$$

where we denote dual norm $\|x\|_* = \sup_{\|x'\| \leq 1} (\langle x, x' \rangle)$.

**Assumption A.3** (Norm Equivalence). As Norm Equivalence itself always holds in finite dimensional spaces, we denote the positive constant, which connects the norm from $\mathcal{X}$ with Euclidean one as $\rho > 0$:

$$\|x\|_* \leq \rho\|x\|_2 \text{ for all } x \in \mathcal{X}$$

Then, we introduce *Stochastic Non-Euclidean Trust-Region Gradient Method with Momentum* version with gradient delay, formulated in Algorithm 3.

---

**Algorithm 3** Stochastic Non-Euclidean Trust-Region Gradient Method with Momentum with Delayed Gradients

1: **input:** $x_0, m_0 \in \mathcal{X}$
2: **parameters:** stepsize $\eta > 0$, momentum $\alpha \in (0, 1)$, number of iterations $K \in \{1, 2, \ldots\}$
3: **for** $k = 0, 1, \ldots, K - 1$ **do**
4:      Sample $\xi_k \sim \mathcal{D}$
5:      $m_{k+1} = (1 - \alpha)m_k + \alpha g(x_{\text{prev(k)}}; \xi_{\text{prev}(k)})$
6:      **if** Standard Async **then**
7:          $x_{k+1} = \underset{\|x - x_k\| \leq \eta}{\arg\min} [< m_{k+1}, x > + R(x)]$
8:      **else if** Error-Feedback (Section 3.2) **then**
9:          $x_{k+1} = x_{\text{prev}(k)} - x_k + 2 \underset{\|x - x_k\| \leq \eta}{\arg\min} \left[ \langle m_{k+1}, x \rangle + R(x) \right] - \underset{\|x - x_{\text{prev}(k)}\| \leq \eta}{\arg\min} \left[ \langle m_{\text{prev}(k)+1}, x \rangle + R(x) \right]$
10:     **end if**
11: **end for**
12: **output:** $x_K \in \mathcal{X}$

---

Here, we denote $\text{prev}(k) = k - \tau$ for an arbitrary delay $\tau > 0$. Additionally, we make a remark that the proposed algorithm matches Muon with $R(\cdot) \equiv 0$ and $\|\cdot\| \equiv \|\cdot\|_{op}$.

**Theorem A.4** (Delayed Muon). *Let Assumptions A.1 - A.3 hold, and let $x_0 \in \text{dom} R$ and $m_0 = g(x_0, \xi_0)$. Then the iterations of Algorithm 3 satisfy the following inequality:*

$$\mathbb{E} \min_{k=1,\ldots,K} \|\nabla f(x_k) + \hat{\nabla} R_k\|_* \leq \frac{\Delta_0}{\eta K} + \frac{2\rho\sigma}{\alpha K}$$
$$+ 2\sqrt{2\alpha}\sqrt{\sigma^2\rho^2 + 2(L\eta\tau)^2}$$
$$+ \frac{7L\eta}{2} + \frac{2L\eta}{\alpha},$$

*where $\hat{\nabla} R_k \in \partial R(x_k)$, $\Delta_0 = F(x_0) - \inf_x F(x)$.*

*Proof.* Our proof extends the convergence framework of (Kovalev, 2025) to accommodate arbitrary gradient delays $\tau \geq 1$, establishing delay-dependent bounds that explicitly capture how staleness propagates through the momentum accumulation process. A detailed version can be found in Appendix B.1. □

**Assumption A.5** (Star Convexity). We assume $f(x)$ to be star-convex:

$$f(\beta x^* + (1 - \beta)x) \leq \beta f(x^*) + (1 - \beta)f(x)$$

for all $x \in \mathcal{X}$, where $\beta \in (0, 1)$.

**Theorem A.6** (Delayed Muon with Weight Decay). *Let Assumptions A.1, A.2 A.3 and A.5 hold, and let $x_0 \in \mathrm{dom}R$ and $m_0 = g(x_0, \xi_0)$. Then the iterations of Algorithm 3 with Weight Decay $\beta > 0$ satisfy the following inequality:*

$$\mathbb{E}[F(x_K) - F(x^*)] \leq (1 - \beta)^K (F(x_0) - F(x^*))$$
$$+ 2\eta \left( \frac{\rho\sigma}{\alpha} + \frac{\sqrt{2\alpha}\sqrt{\sigma^2\rho^2 + 8(L\eta\tau)^2}}{\beta} \right)$$
$$+ \frac{4L\eta^2}{\beta} \left( 1 + \frac{1}{\alpha} \right).$$

*where $\eta$ and $\beta$ satisfy the following:*

$$\eta \geq \beta \max \{\|x_0\|, \|x^*\|\} \tag{5}$$

*Proof.* We establish this result by integrating our delay-aware convergence framework from Theorem A.4 with the weight decay analysis methodology of (Kovalev, 2025). A detailed version can be found in Appendix B.4. □

**Discussion**. The main change comparing the obtained estimation for delayed setup with the synchronous one is in the noise bound, which previously occurred only from the stochastic oracle noise term and now it's enlarged due to gradient delay

## B. Proofs for the General Theory

### B.1. Proof of Theorem A.4

We first start with the formulating of the *descent lemma* from (Kovalev, 2025). We highlight that it stays true in the delayed setup, since the delay affects the momentum terms.

**Lemma B.1.** *Let Assumption A.2 hold, and let $x_0 \in \mathrm{dom}R$. Then the iterations of Algorithm 3 satisfy the following inequality:*

$$F(x_{k+1}) \leq F(x_k) - \eta\|\nabla f(x_{k+1}) + \hat{\nabla}R_{k+1}\|_* + 2\eta\|\nabla f(x_{k+1}) - m_{k+1}\|_* + \frac{3}{2}L\eta^2, \tag{6}$$

*where $\hat{\nabla}R_{k+1} \in \partial R(x_{k+1})$.*

Next, we establish a key lemma that bounds the momentum's tracking error.

**Lemma B.2.** *Let Assumptions A.2, A.1, A.3 hold, and let $x_0 \in \mathrm{dom}R$ and $m_0 = g(x_0, \xi_0)$. Then the iterations of Algorithm 3 satisfy the following inequality for $k \geq 0$:*

$$\mathbb{E}\|m_{k+1} - \nabla f(x_k)\|_* \leq (1 - \alpha)^{k+1}\rho\sigma + \sqrt{2\alpha}\sqrt{\sigma^2\rho^2 + 2(L\eta\tau)^2} + \frac{L\eta}{\alpha}. \tag{7}$$

Using Lemma B.1, we obtain the following inequality:

$$\min_{k=1,\ldots,K} \|\nabla f(x_k) + \hat{\nabla}R_k\|_* \leq \frac{F(x_0) - \inf_x F(x)}{\eta K} + \frac{3L\eta}{2} + \frac{2}{K}\sum_{k=1}^{K}\|\nabla f(x_k) - m_k\|_*$$
$$\leq \frac{F(x_0) - \inf_x F(x)}{\eta K} + \frac{7L\eta}{2} + \frac{2}{K}\sum_{k=0}^{K-1}\|\nabla f(x_k) - m_{k+1}\|_*,$$

Using Lemma B.2, we obtain

$$\mathbb{E} \min_{k=1,\dots,K} \|\nabla f(x_k) + \hat{\nabla} R_k\|_* \leq \frac{F(x_0) - \inf_x F(x)}{\eta K} + \frac{7L\eta}{2} + \frac{2L\eta}{\alpha} + \frac{2\rho\sigma}{\alpha K} + 2\sqrt{2\alpha}\sqrt{\sigma^2\rho^2 + 2(L\eta\tau)^2}.$$

$\square$

## B.2. Proof of Lemma B.2

We can express $m_{k+1} - \nabla f(x_k)$ as follows using $m_{k+1}$ definition in Algorithm 3:

$$\begin{aligned}
m_{k+1} - \nabla f(x_k) &= (1-\alpha)m_k + \alpha g(x_{\text{prev}(k)}; \xi_{\text{prev}(k)}) - \nabla f(x_k) \\
&= (1-\alpha)(m_k - \nabla f(x_{k-1})) + \alpha(g(x_{\text{prev}(k)}; \xi_{\text{prev}(k)}) - \nabla f(x_k)) \\
&\quad + (1-\alpha)(\nabla f(x_{k-1}) - \nabla f(x_k)).
\end{aligned}$$

This implies the following for all $k \geq 0$:

$$m_{k+1} - \nabla f(x_k) = (1-\alpha)^{k+1}(m_0 - \nabla f(x_0)) + \sum_{i=0}^{k-1}(1-\alpha)^{k-i}(\nabla f(x_i) - \nabla f(x_{i+1}))$$

$$+ \sum_{i=0}^{k}\alpha(1-\alpha)^{k-i}(g(x_{\text{prev}(k)}; \xi_{\text{prev}(k)}) - \nabla f(x_i)).$$

Using this, we can upper-bound $\mathbb{E}\|m_{k+1} - \nabla f(x_k)\|_*$ for $k \geq 0$ as follows:

$$\begin{aligned}
\mathbb{E}\|m_{k+1} - \nabla f(x_k)\|_* &\overset{(a)}{\leq} (1-\alpha)^{k+1}\mathbb{E}\|m_0 - \nabla f(x_0)\|_* \\
&\quad + \sum_{i=0}^{k-1}(1-\alpha)^{k-i}\|\nabla f(x_i) - \nabla f(x_{i+1})\|_* \\
&\quad + \mathbb{E}\left\|\sum_{i=0}^{k}\alpha(1-\alpha)^{k-i}(g(x_{\text{prev}(i)}, \xi_{\text{prev}(i)}) - \nabla f(x_i))\right\|_* \\
&\overset{(b)}{\leq} (1-\alpha)^{k+1}\mathbb{E}\|m_0 - \nabla f(x_0)\|_* + \sum_{i=0}^{k-1}(1-\alpha)^{k-i}L\eta \\
&\quad + \mathbb{E}\left\|\sum_{i=0}^{k}\alpha(1-\alpha)^{k-i}(g(x_{\text{prev}(i)}, \xi_{\text{prev}(i)}) - \nabla f(x_i))\right\|_* \\
&\overset{(c)}{\leq} (1-\alpha)^{k+1}\rho\mathbb{E}\|m_0 - \nabla f(x_0)\|_2 + \sum_{i=0}^{k-1}(1-\alpha)^{k-i}L\eta \\
&\quad + \mathbb{E}\left\|\sum_{i=0}^{k}\alpha(1-\alpha)^{k-i}(g(x_{\text{prev}(i)}, \xi_{\text{prev}(i)}) - \nabla f(x_i))\right\|_* \\
&\overset{(d)}{\leq} (1-\alpha)^{k+1}\rho\sqrt{\mathbb{E}\|m_0 - \nabla f(x_0)\|_2^2} + \sum_{i=0}^{k-1}(1-\alpha)^{k-i}L\eta \\
&\quad + \sqrt{\mathbb{E}\left\|\sum_{i=0}^{k}\alpha(1-\alpha)^{k-i}(g(x_{\text{prev}(i)}, \xi_{\text{prev}(i)}) - \nabla f(x_i))\right\|_*^2},
\end{aligned}$$

where (a) expand the momentum update rule and apply triangle inequality; (b) use $L$-smoothness of $f$ and the constraint $\|x_{i+1} - x_i\| \leq \eta$; (c) use the norm compatibility property $\|\cdot\|_* \leq \rho\|\cdot\|_2$, we keep the dual norm for the second term; (d) apply Jensen's inequality $\mathbb{E}[\|X\|] \leq \sqrt{\mathbb{E}[\|X\|^2]}$.

Now, we focus on the delayed-gradient term estimation. We first split the error into stochastic-noise and drift components:

$$\sum_{i=0}^{k} \alpha(1-\alpha)^{k-i}\big(g(x_{\mathrm{prev}(i)};\xi_{\mathrm{prev}(i)}) - \nabla f(x_i)\big)$$

$$= \underbrace{\sum_{i=0}^{k} \alpha(1-\alpha)^{k-i}\big(g(x_{\mathrm{prev}(i)};\xi_{\mathrm{prev}(i)}) - \nabla f(x_{\mathrm{prev}(i)})\big)}_{S_1} + \underbrace{\sum_{i=0}^{k} \alpha(1-\alpha)^{k-i}\big(\nabla f(x_{\mathrm{prev}(i)}) - \nabla f(x_i)\big)}_{S_2}. \quad (8)$$

Using $\|S_1 + S_2\|_*^2 \le 2\|S_1\|_*^2 + 2\|S_2\|_*^2$, we bound the two terms separately. For $S_1$, Assumption A.3 gives $\mathbb{E}\|S_1\|_*^2 \le \rho^2\mathbb{E}\|S_1\|_2^2$. When expanding the Euclidean square, the cross terms vanish in expectation by conditional unbiasedness: although the iterates depend on past samples, the noise $g(x_j;\xi_j) - \nabla f(x_j)$ has zero conditional mean given the previous randomness, while earlier noise terms are already determined. Thus, by Assumption A.1,

$$\mathbb{E}\|S_1\|_*^2 \le \rho^2 \sum_{i=0}^{k} \alpha^2(1-\alpha)^{2(k-i)}\mathbb{E}\left\|g(x_{\mathrm{prev}(i)};\xi_{\mathrm{prev}(i)}) - \nabla f(x_{\mathrm{prev}(i)})\right\|_2^2$$

$$\le \rho^2\sigma^2 \sum_{i=0}^{k} \alpha^2(1-\alpha)^{2(k-i)}. \quad (9)$$

For $S_2$, no cancellation is used. Instead, by Assumption A.2 and the delay bound,

$$\|\nabla f(x_{\mathrm{prev}(i)}) - \nabla f(x_i)\|_* \le L\|x_{\mathrm{prev}(i)} - x_i\| \le L\eta\tau. \quad (10)$$

Combining these estimates, we obtain

$$\mathbb{E}\left\|\sum_{i=0}^{k} \alpha(1-\alpha)^{k-i}\big(g(x_{\mathrm{prev}(i)};\xi_{\mathrm{prev}(i)}) - \nabla f(x_i)\big)\right\|_*^2$$

$$\le 2\sum_{i=0}^{k} \alpha^2(1-\alpha)^{2(k-i)}\left(\rho^2\sigma^2 + 2(L\eta\tau)^2\right). \quad (11)$$

Therefore, continuing our derivations, we easily obtain

$$\le (1-\alpha)^{k+1}\rho\sigma + \sum_{i=0}^{k-1}(1-\alpha)^{k-i}L\eta + \sqrt{2\alpha}\sqrt{\sigma^2\rho^2 + 2(L\eta\tau)^2}$$

$$\le (1-\alpha)^{k+1}\rho\sigma + \frac{L\eta}{\alpha} + \sqrt{2\alpha}\sqrt{\sigma^2\rho^2 + 2(L\eta\tau)^2}$$

$\square$

## B.3. Proof of Error-Feedback convergence

**Theorem B.3** (EF Delayed Muon). *Let Assumptions A.1 - A.3 hold, and let $x_0 \in \mathrm{dom}R$, while $R \equiv 0$ and $m_0 = g(x_0, \xi_0)$. Then the iterations of Algorithm 3 satisfy the following inequalities:*

$$\mathbb{E}\min_{k=1,\dots,K}\|\nabla f(x_k)\|_* \le \frac{\Delta_0}{\eta K} + \frac{(2\tau+2)\rho\sigma}{\alpha K}$$

$$+ (2\tau+2)\sqrt{2\alpha}\sqrt{\rho^2\sigma^2 + 2(2\tau+1)^2(L\eta\tau)^2}$$

$$+ \frac{3(2\tau+1)^2 L\eta}{2} + (2\tau+2)(2\tau+1)L\eta + \frac{(2\tau+2)(2\tau+1)L\eta}{\alpha}$$

*Proof.* We start the proof with an extended version of Lemma B.1.

**Lemma B.4.** *Let Assumption A.2 hold, and let $x_0 \in \mathrm{dom} R$ and $R \equiv 0$. Then the iterations of Error-Feedback in Algorithm 3 satisfy the following inequality:*

$$F(x_{k+1}) \leq F(x_k) + \tfrac{3}{2}(2\tau + 1)^2 L\eta^2 + (2\tau + 2)\eta \|\nabla f(x_{k+1}) - m_{k+1}\|_* - \eta \|\nabla f(x_{k+1})\|_* \tag{12}$$

Using Lemma B.4 we obtain the following estimation:

$$\min_{k=1,\ldots,K} \|\nabla f(x_k)\|_* \leq \frac{F(x_0) - \inf_x F(x)}{\eta K} + \frac{3(2\tau + 1)^2 L\eta}{2} + \frac{(2\tau + 2)}{K} \sum_{k=1}^{K} \|\nabla f(x_k) - m_k\|_*$$

$$\leq \frac{F(x_0) - \inf_x F(x)}{\eta K} + \frac{3(2\tau + 1)^2 L\eta}{2} + (2\tau + 2)(2\tau + 1)L\eta + \frac{(2\tau + 2)}{K} \sum_{k=0}^{K-1} \|\nabla f(x_k) - m_{k+1}\|_*,$$

Then using the same results for Delayed momentum version from Lemma B.2 and combining it with $\|x_{k+1} - x_k\| \leq (2\tau + 1)\eta$, we obtain

$$\mathbb{E} \min_{k=1,\ldots,K} \|\nabla f(x_k)\|_* \leq \frac{F(x_0) - \inf_x F(x)}{\eta K} + \frac{3(2\tau + 1)^2 L\eta}{2} + (2\tau + 2)(2\tau + 1)L\eta + \frac{(2\tau + 2)(2\tau + 1)L\eta}{\alpha}$$

$$+ \frac{(2\tau + 2)\rho\sigma}{\alpha K} + (2\tau + 2)\sqrt{2}\sqrt{\alpha}\sqrt{\rho^2\sigma^2 + 2(2\tau + 1)^2(L\eta\tau)^2}.$$

$\square$

### B.3.1. PROOF OF LEMMA B.4

We can upper-bound $F(x_{k+1})$ as follows:

$$
\begin{aligned}
F(x_{k+1}) &\overset{(a)}{=} f(x_{k+1}) + R(x_{k+1}) \\
&\overset{(b)}{\leq} f(x_k) + \langle \nabla f(x_k), x_{k+1} - x_k \rangle + \tfrac{1}{2}L\|x_{k+1} - x_k\|_2^2 + R(x_{k+1}) \\
&\overset{(c)}{=} f(x_k) + \tfrac{1}{2}L\|x_{k+1} - x_k\|_2^2 + R(x_{k+1}) \\
&\quad + \langle m_{k+1} + \nabla f(x_{k+1}) - m_{k+1} + \nabla f(x_k) - \nabla f(x_{k+1}), x_{k+1} - x_k \rangle \\
&\overset{(d)}{\leq} f(x_k) + \tfrac{1}{2}L\|x_{k+1} - x_k\|_2^2 + R(x_{k+1}) + \langle m_{k+1}, x_{k+1} - x_k \rangle \\
&\quad + \|x_{k+1} - x_k\|\|\nabla f(x_{k+1}) - m_{k+1}\|_* + \|x_{k+1} - x_k\|\|\nabla f(x_k) - \nabla f(x_{k+1})\|_* \\
&\overset{(e)}{\leq} f(x_k) + \tfrac{3}{2}L\|x_{k+1} - x_k\|_2^2 + \|x_{k+1} - x_k\|\|\nabla f(x_{k+1}) - m_{k+1}\|_* \\
&\quad + R(x_{k+1}) + \langle m_{k+1}, x_{k+1} - x_k \rangle,
\end{aligned}
$$

where (a) use the definition of function $F(x)$; (b) and (e) use Assumption A.2; (c) algebraic manipulation: add and subtract $m_{k+1}$ and $\nabla f(x_{k+1})$; (d) use the definition of dual norm.

Then we develop an estimation for $R(x_{k+1}) + \langle m_{k+1}, x_{k+1} - x_k \rangle$.

Using Error-Feedback from Algorithm 3,

$$x_{k+1} = \tau x_{\mathrm{prev}(k)} - \tau x_k + (\tau + 1) \arg\min_{\|x - x_k\| \leq \eta} [\langle m_{k+1}, x \rangle + R(x)] - \tau \arg\min_{\|x - x_p\| \leq \eta} [\langle m_{\mathrm{prev}(k)+1}, x \rangle + R(x)]$$

Thus, we obtain

$$R(x_{k+1}) + \left\langle m_{k+1}, \tau x_{\mathrm{prev}(k)} - (\tau + 1)x_k + (\tau + 1) \arg\min_{\|x - x_k\| \leq \eta} [\langle m_{k+1}, x \rangle + R(x)] - \tau \arg\min_{\|x - x_p\| \leq \eta} [\langle m_{\mathrm{prev}(k)+1}, x \rangle + R(x)] \right\rangle$$

$$= R(x_{k+1}) + \tau \left\langle m_{k+1}, x_{\mathrm{prev}(k)} - \arg\min_{\|x - x_p\| \leq \eta} [\langle m_{\mathrm{prev}(k)+1}, x \rangle + R(x)] \right\rangle - (\tau + 1) \left\langle m_{k+1}, x_k - \arg\min_{\|x - x_k\| \leq \eta} [\langle m_{k+1}, x \rangle + R(x)] \right\rangle$$

Then we apply Lemma 3 from (Kovalev, 2025) to the first two terms:

$$\leq (\tau + 1) \cdot \left( R(x_k) - \eta \| m_{k+1} + \hat{\nabla} R_{k+1} \|_* \right)$$
$$- \tau \left( R(x_{k+1}) - \langle m_{k+1}, \arg \min_{\|x - x_p\| \leq \eta} (\langle m_{\mathrm{prev}(k)+1}, x \rangle + R(x)) - x_{\mathrm{prev}(k)} \rangle \right)$$

Next, we apply $R \equiv 0$ and Cauchy-Schwarz inequality to the third term and obtain:

$$\leq -(\tau + 1)\eta \| m_{k+1} \|_* + \tau \eta \| m_{k+1} \|_* = -\eta \| m_{k+1} \|_*$$

Then we estimate $\|x_{k+1} - x_k\|$ using the EF update from Algorithm 3.

$$\|x_{k+1} - x_k\| = \left\| (\tau + 1) x_k - \tau x_{\mathrm{prev}(k)} - (\tau + 1) \arg \min_{\|x - x_k\| \leq \eta} [\langle m_{k+1}, x \rangle + R(x)] + \tau \arg \min_{\|x - x_{\mathrm{prev}(k)}\| \leq \eta} [\langle m_{\mathrm{prev}(k)+1}, x \rangle + R(x)] \right\|$$

$$\leq (\tau + 1) \left\| x_k - \arg \min_{\|x - x_k\| \leq \eta} [\langle m_{k+1}, x \rangle + R(x)] \right\| + \tau \left\| x_{\mathrm{prev}(k)} - \arg \min_{\|x - x_{\mathrm{prev}(k)}\| \leq \eta} [\langle m_{\mathrm{prev}(k)+1}, x \rangle + R(x)] \right\|$$

$$\leq (2\tau + 1)\eta$$

Continuing estimation we obtain

$$\leq f(x_k) + \tfrac{3}{2} L \| x_{k+1} - x_k \|_2^2 + \| x_{k+1} - x_k \| \| \nabla f(x_{k+1}) - m_{k+1} \|_*$$
$$- \eta \| m_{k+1} \|_*$$
$$\leq f(x_k) + \tfrac{3}{2} (2\tau + 1)^2 L \eta^2 + (2\tau + 1)\eta \| \nabla f(x_{k+1}) - m_{k+1} \|_*$$
$$- \eta \| m_{k+1} \|_*$$
$$= F(x_k) + \tfrac{3}{2} (2\tau + 1)^2 L \eta^2 + (2\tau + 1)\eta \| \nabla f(x_{k+1}) - m_{k+1} \|_* - \eta \| m_{k+1} \|_*$$
$$\leq F(x_k) + \tfrac{3}{2} (2\tau + 1)^2 L \eta^2 + (2\tau + 2)\eta \| \nabla f(x_{k+1}) - m_{k+1} \|_* - \eta \| \nabla f(x_{k+1}) \|_*$$

$\square$

## B.4. Proof of Theorem A.6

In this proof, we are going to use Lemma B.5 from (Kovalev, 2025).

**Lemma B.5.** *Under the conditions of Equation 5, let $x \in \mathcal{X}$ be defined as follows:*

$$x = \beta x^* + (1 - \beta) x_k. \tag{13}$$

*Then, the following inequalities hold:*

$$\|x - (1 - \beta)x_k\| \leq \eta, \quad \|x - x_k\| \leq 2\eta, \quad \|x - x_{k+1}\| \leq 2\eta, \quad \|x_{k+1} - x_k\| \leq 2\eta. \tag{14}$$

Additionally, we obtain the following Lemma B.6.

**Lemma B.6.** *Let Assumptions A.1 - A.3 hold, and let $x_0 \in \mathrm{dom} R$ and $m_0 = g(x_0, \xi_0)$. Then the iterations of Algorithm 3 with Weight Decay satisfy the following inequality for $k \geq 0$:*

$$\mathbb{E}[\| m_{k+1} - \nabla f(x_k) \|_*] \leq (1 - \alpha)^{k+1} \rho \sigma + \sqrt{2\alpha} \sqrt{\rho^2 \sigma^2 + 8(L\eta\tau)^2} + \frac{2L\eta}{\alpha}. \tag{15}$$

The proof is similar to Section B.2, with $\|x_{k+1} - x_k\| \leq 2\eta$.

The proof for Theorem A.6 is similar to Theorem 4 from (Kovalev, 2025); we obtain

$$\mathbb{E}[F(x_{k+1}) - F(x^*)] \leq (1 - \beta)\mathbb{E}[F(x_k) - F(x^*)] + 2\eta\rho\sigma(1 - \alpha)^k + 2\eta\sqrt{2\alpha}\sqrt{\rho^2 \sigma^2 + 8(L\eta\tau)^2}$$
$$+ 4L\eta^2 + \frac{4L\eta^2}{\alpha},$$

which implies the following inequality:

$$\mathbb{E}[F(x_K) - F(x^*)] \leq (1 - \beta)^K (F(x_0) - F(x^*)) + 2\eta \left( \frac{\rho\sigma}{\alpha} + \frac{\sqrt{2\alpha}\sqrt{\rho^2\sigma^2 + 8(L\eta\tau)^2}}{\beta} \right) + \frac{4L\eta^2}{\beta} \left( 1 + \frac{1}{\alpha} \right).$$

$\square$

## C. Experimental Setup

We focus on the standard next-token prediction task, training decoder-only models based on the SmolLM2 architecture (Allal et al., 2025) with parameter counts of 135M and 360M using the Fineweb-Edu dataset (Penedo et al., 2024). Unless explicitly stated otherwise, all training runs adhere to a Chinchilla compute-optimal token-to-parameter ratio of 20:1 (Hoffmann et al., 2022).

### C.1. Hyperparameters and Training Details

To ensure a rigorous and fair comparison across different optimization algorithms, we adopted a systematic approach to hyperparameter tuning, prioritizing the stability and optimality of the synchronous baselines.

**Optimizer Tuning.** We began by establishing strong baselines for the SmolLM-2 models (Allal et al., 2025) (135M and 360M) using AdamW (Loshchilov & Hutter, 2017). We performed a grid search over learning rates and weight decay values, using a multiplicative step of 2 (uniform grid in log scale) for the learning rate and testing four distinct weight decay values. After verifying that a weight decay of 0.1 consistently yielded optimal results, we fixed this value for the remainder of the study. The resulting optimal learning rates were found to be 4e-3 for the 135M model and 2e-3 for the 360M model.

For all other optimizers (with the exception of Lion (Chen et al., 2023), which operates on a distinct scale), we tuned the learning rate within a narrow range surrounding the optimal AdamW values. This approach leverages prior findings (Wen et al., 2025; Semenov et al., 2025) indicating that optimal hyperparameters for many modern optimizers tend to cluster in similar regions. Optimal weight decay for Lion was 0.5 and learning rate was approximately 5e-4, which aligns with results from Wen et al. (2025). Crucially, we always compared synchronous and asynchronous runs using *identical* hyperparameter configurations.

**Batch Size Selection.** We aimed to approximate optimal batch sizes for valid scaling laws. For the 135M and 360M models, we selected global batch sizes based on the average of predictions derived from Li et al. (2025b) and Bi et al. (2024). Although the context length was set to 1024, the use of padding for the FineWeb dataset resulted in an average sequence length of approximately ~700 tokens. Consequently, a global batch size of 256 for the 135M model and 512 for the 360M model resulted in effective batch sizes of approximately 180K and 360K tokens, respectively.

For the larger 2B and 10B MoE models, we utilized batch sizes slightly larger than theoretical optima to maximize GPU utilization. For 2B training we used 1M, 1.5M and 2.25M tokens batch sizes for 50B, 100B, and 200B respectively, and for 10B model training on 200B tokens we used 4M batch size. It is important to note that this regime theoretically disadvantages Async PP: larger batch sizes imply fewer total optimization steps for the same token budget, leaving the model with fewer opportunities to recover from the initial errors caused by gradient delays Figure 1. Thus, the robustness observed in our large-scale experiments is likely a conservative estimate.

**Other Settings.** For all experiments, we utilized a cosine decay learning rate schedule with a minimum learning rate of $0.1 \times \text{max\_lr}$. We used 10% of Chinchilla tokens for learning rate warmup. We also explored varying gradient clipping thresholds but observed no significant impact on performance for either synchronous or asynchronous runs; therefore, standard clipping value of 1.0 was maintained.

### C.2. Model architectures

In our experiments, we utilize four distinct model architectures: two dense models from the SmolLM-2 family (135M and 360M parameters) and two custom sparse Mixture-of-Experts (MoE) models with 2B and 10B total parameters.

**SmolLM-2 Models.** We employ the SmolLM-2 (Allal et al., 2025) 135M and 360M architectures as our dense baselines. Built upon the standard Llama architecture, these models incorporate Grouped Query Attention (Ainslie et al., 2023), RMSNorm (Zhang & Sennrich, 2019) and SwiGLU (Shazeer, 2020). Both models are trained with a context length of 1,024

tokens and a vocabulary size of 49,152.

**Custom MoE Models.** To rigorously validate our hypotheses at scale, we trained two custom MoE models. Both models utilize a tokenizer with a vocabulary size of 128k and support a context length of 8,192 tokens.

- **2B MoE (0.5B Active):** This model features 16 layers with a hidden size of 1024. It uses 16 query heads and 4 key-value heads. The routing mechanism involves 64 experts with top-8 gating. Despite the total parameter count of $\approx$2B, the active parameter count per token is approximately 500M.

- **10B MoE (0.65B Active):** This model employs a hybrid architecture inspired by QwenTeam (2025), incorporating Gated DeltaNet (Yang et al., 2024) layers. It consists of 24 layers, configured such that every 4th layer uses Full Attention while the remaining layers utilize linear attention. The model scales to 512 experts with top-10 gating and utilizes a Shared Expert and Shared Expert Trainable Weight mechanism. While the total parameter count is $\approx$10B, the highly sparse architecture maintains an efficient active parameter count of only $\approx$0.65B during inference.

## D. Additional Experiments

In this section, we present supplementary experiments that were omitted from the main text due to space constraints, along with extended analyses for the 135M and 360M model configurations.

### D.1. AdamW Ablations

To better understand why AdamW degrades more severely than other optimizers under one-step delay, we perform several targeted diagnostic experiments on the 135M model. These experiments do not fully isolate a single cause of AdamW's degradation, but they help rule out several simple explanations and provide additional evidence for the importance of first-moment dynamics.

We first compare stale updates with the corresponding fresh updates that would have been applied in a non-delayed run. Specifically, we measure update cosine similarity and relative update error between these two updates; see Figures 6a, 6b, 7a and 7b. Somewhat surprisingly, these discrepancy metrics are not worse for AdamW than for Muon. Thus, AdamW's poor delayed performance cannot be explained simply by its stale updates being more different from their fresh counterparts according to these direct update-level metrics.

We next test whether the degradation is driven by the final language-model head, which has been identified as a sensitive component in optimizer studies (Zhao et al., 2025). To do so, we keep the LM head synchronous while applying delayed updates to the rest of the model. As shown in Table 5, this modification provides little improvement: AdamW still remains far worse than its synchronous baseline, with final loss above 3.0 in the 135M setting. This suggests that the instability is not localized to the LM head.

Finally, we isolate the effect of delaying different AdamW state variables. When the delay is applied only to the first-moment update $m_t$, while the second-moment update $v_t$ remains synchronous, the resulting loss is almost identical to fully delayed AdamW (see Table 5). This supports the interpretation in Section 2.2 that first-moment dynamics play a central role in robustness to one-step delay.

A likely reason is that the value of $\beta_1$ required for delay robustness may fall outside the stable region for AdamW itself. In our experiments, increasing $\beta_1$ improves delayed robustness only up to a point, while very large values destabilize or degrade AdamW even in the synchronous setting. For example, synchronous AdamW with $\beta_1 = 0.99$ reaches a substantially worse final loss than the standard $\beta_1 = 0.9$ configuration (2.939 vs. 2.877). This contrasts with optimizers such as SOAP or Adan, whose stable operating regimes include substantially larger first-moment coefficients.

| Setup | Sync | EF | Async |
|---|---|---|---|
| AdamW, $\beta = (0.9, 0.95)$ | 2.879 | 2.920 | 3.158 |
| AdamW, `no_delay_lmhead`, $\beta = (0.9, 0.95)$ | 2.879 | 2.917 | **3.100** |
| AdamW, $\beta = (0.95, 0.95)$ | 2.877 | 2.901 | 3.227 |
| AdamW, $\beta = (0.9, 0.99)$ | 2.876 | 2.897 | – |
| Adam delay $m$, $\beta = (0.9, 0.99)$ | 2.875 | 2.898 | 3.190 |
| AdamW, $\beta = (0.95, 0.99)$ | 2.875 | **2.895** | – |
| Adam delay $m$, $\beta = (0.95, 0.99)$ | 2.876 | 2.896 | 3.450 |
| AdamW, $\beta = (0.95, 0.999)$ | **2.873** | 2.908 | 3.241 |
| AdamW, `no_delay_lmhead`, $\beta = (0.95, 0.999)$ | **2.873** | 2.922 | 3.289 |

**Table 5.** Diagnostic AdamW ablations on the 135M model. We compare fully delayed AdamW, AdamW with only the first-moment update delayed, and AdamW with a synchronous LM head. The results suggest that delaying the first-moment dynamics closely reproduces the behavior of fully delayed AdamW, while keeping the LM head synchronous does not remove the degradation.

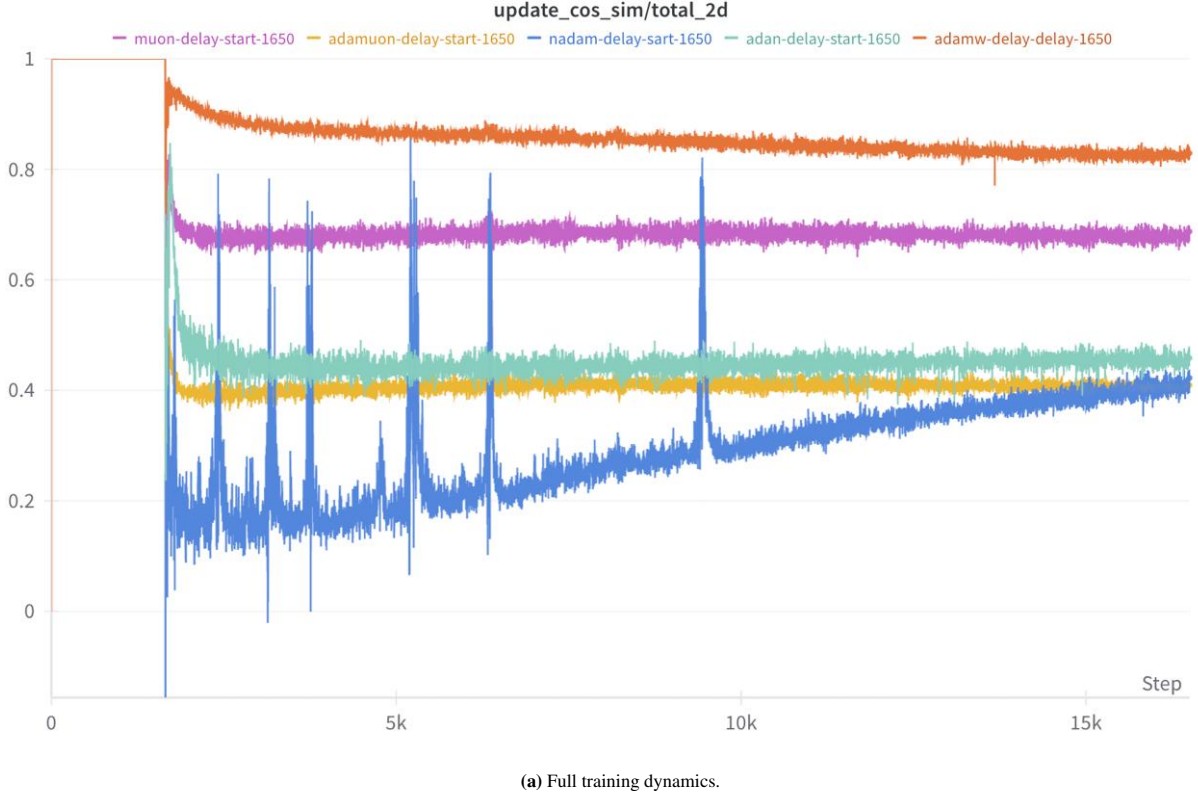

**(a)** Full training dynamics.

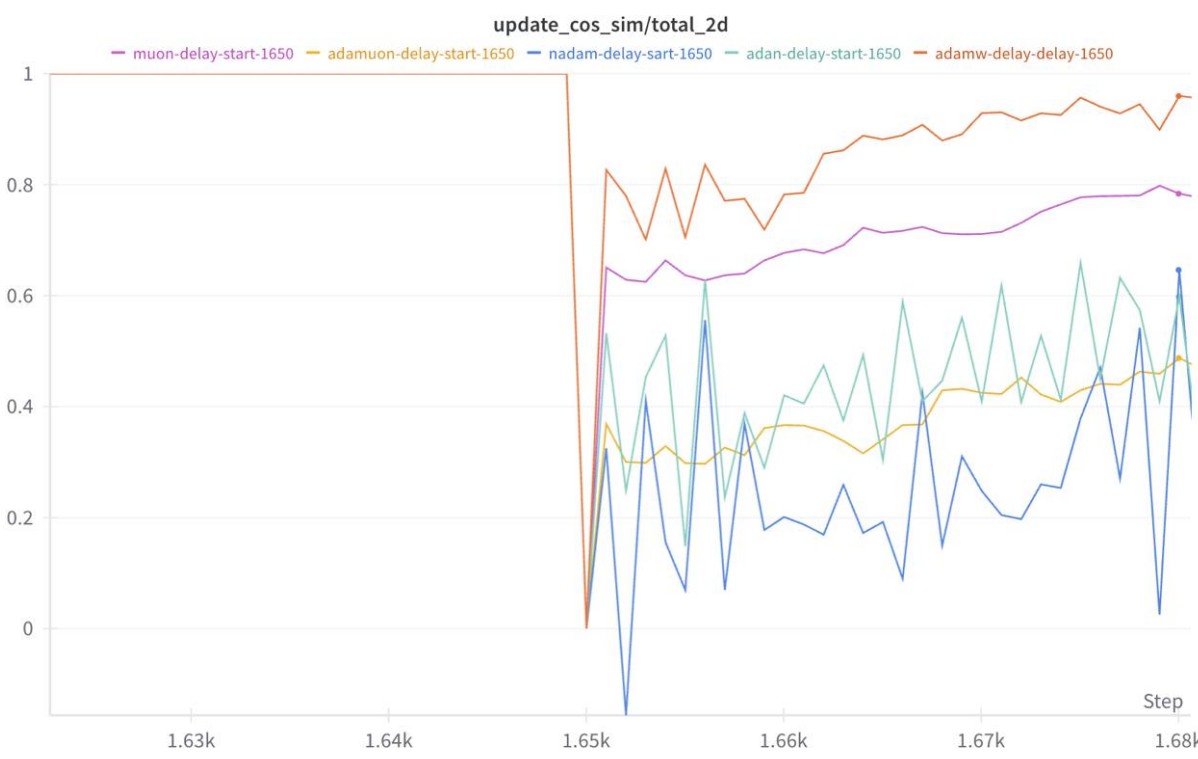

**(b)** Zoomed-in view around delay start, showing the detailed behavior.

**Figure 6.** Cosine similarity between the delayed optimizer update and the corresponding fresh update on the 135M model. The fresh update is defined as the update that would have been applied using the non-delayed gradient at the same step. The bottom panel zooms in on the iterations around the transition to one-step delayed training.

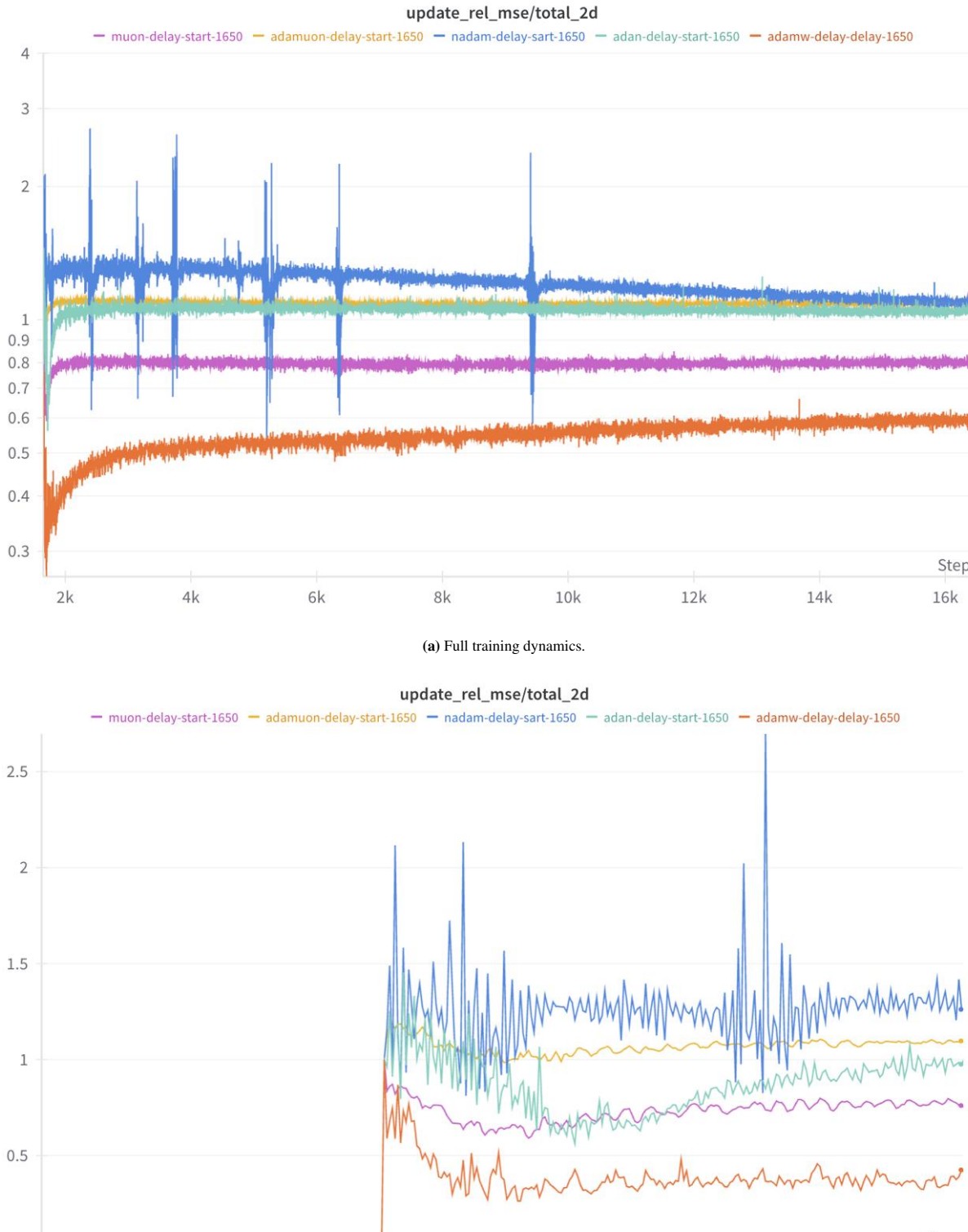

**(a)** Full training dynamics.

**(b)** Zoomed-in view around delay start, showing the detailed behavior.

**Figure 7.** Relative update error between the delayed optimizer update and the corresponding fresh update on the 135M model. The fresh update is defined as the update that would have been applied using the non-delayed gradient at the same step. The bottom panel zooms in on the iterations around the transition to one-step delayed training.

## D.2. Comparison with SAPipe-Style Weight Prediction

SAPipe (Chen et al., 2022) studies staleness-aware training in a data-parallel setting, where one-step staleness arises from delayed gradient aggregation across workers. Although this staleness comes from a different systems mechanism than Async PP, both settings consider optimization with a one-step delayed update. SAPipe mitigates this delay using weight prediction, whereas our main method uses an update-level Error Feedback correction. We therefore include a small-scale comparison between these two mitigation strategies on the 135M model.

The results are shown in Table 6. SAPipe-style weight prediction performs on par with Error Feedback across the tested optimizers, matching it exactly for Muon and remaining very close for AdamW and SOAP. This indicates that weight prediction is another effective strategy for mitigating one-step staleness. Since we became aware of SAPipe only after completing the main experimental study, we leave a more extensive evaluation of SAPipe-style corrections in Async PP to future work.

| Optimizer | Synchronous Baseline | Error Feedback (ours) | SAPipe | Asynchronous |
|-----------|----------------------|-----------------------|--------|--------------|
| Muon | 2.839 | **2.845** | **2.845** | 2.856 |
| AdamW | 2.877 | **2.901** | 2.902 | 3.227 |
| SOAP | 2.850 | **2.858** | 2.862 | 2.872 |

**Table 6.** Comparison of our update-level Error Feedback with a SAPipe-style weight-prediction correction on the 135M model. Both methods substantially reduce the gap between synchronous and standard asynchronous training. SAPipe-style weight prediction performs on par with Error Feedback, suggesting that weight prediction is another effective mitigation strategy for one-step staleness.

## D.3. Ablation on Synchronous Cooldown

As discussed in Section 3.1, we investigated a "synchronous cooldown" strategy, where the training process switches from asynchronous to synchronous mode towards the end of training. The hypothesis was that removing stale gradients in the final convergence phase might recover the remaining performance gap. We conducted ablation studies on the 135M model for both Muon and AdamW. For AdamW, we utilized $\beta_1 = 0.95$, synchronous warmup of $1W$, and Error-Feedback enabled, while for Muon we used the standard configuration with async start at step 0, no EF. The switch-over point was defined relative to the warmup duration $W$ (e.g., $-1.5W$ indicates switching to synchronous mode $1.5 \times W$ steps before the end of training).

The results are summarized in Table 7. We observe that switching back to synchronous training yields only marginal improvements.

**Table 7.** Ablation study on switching to synchronous training near the end of the schedule (135M model). Cutoff times are expressed relative to the warmup duration $W$.

| | Sync | No Switch | Switch to Sync (Time before end) | | | |
|---|------|-----------|---------|---------|---------|---------|
| Configuration | Baseline | (Async throughout) | $-1.5W$ | $-1.0W$ | $-0.5W$ | $-0.25W$ |
| Muon (start=0, no EF) | 2.839 | 2.856 | 2.853 | 2.853 | 2.853 | 2.853 |
| AdamW ($\beta_1 = 0.95$, start=$1W$, EF) | 2.879 | 2.935 | 2.928 | 2.928 | 2.930 | 2.929 |

## D.4. DC-ASGD Delay Compensation

We also evaluate Delay-Compensated ASGD (DC-ASGD) (Zheng et al., 2017), a gradient-level staleness correction based on a Taylor-style compensation term. Because the correction magnitude is controlled by the coefficient $\lambda$, we sweep $\lambda$ from $10^4$ to $10^8$ on SmoLLM-135M with Muon. As shown in Figure 8, none of the tested coefficients improves over the standard delayed baseline. Small values of $\lambda$ leave the result essentially unchanged, while larger values degrade training.

## D.5. Gradient-Based Error Feedback

We also evaluate a gradient-level variant of Error Feedback, where the correction is applied to raw gradients before they are passed to the optimizer. As shown in Figure 9, this variant is unstable and diverges in our experiments. We do not investigate the cause of this instability in detail, and use the update-level correction throughout the main experiments.

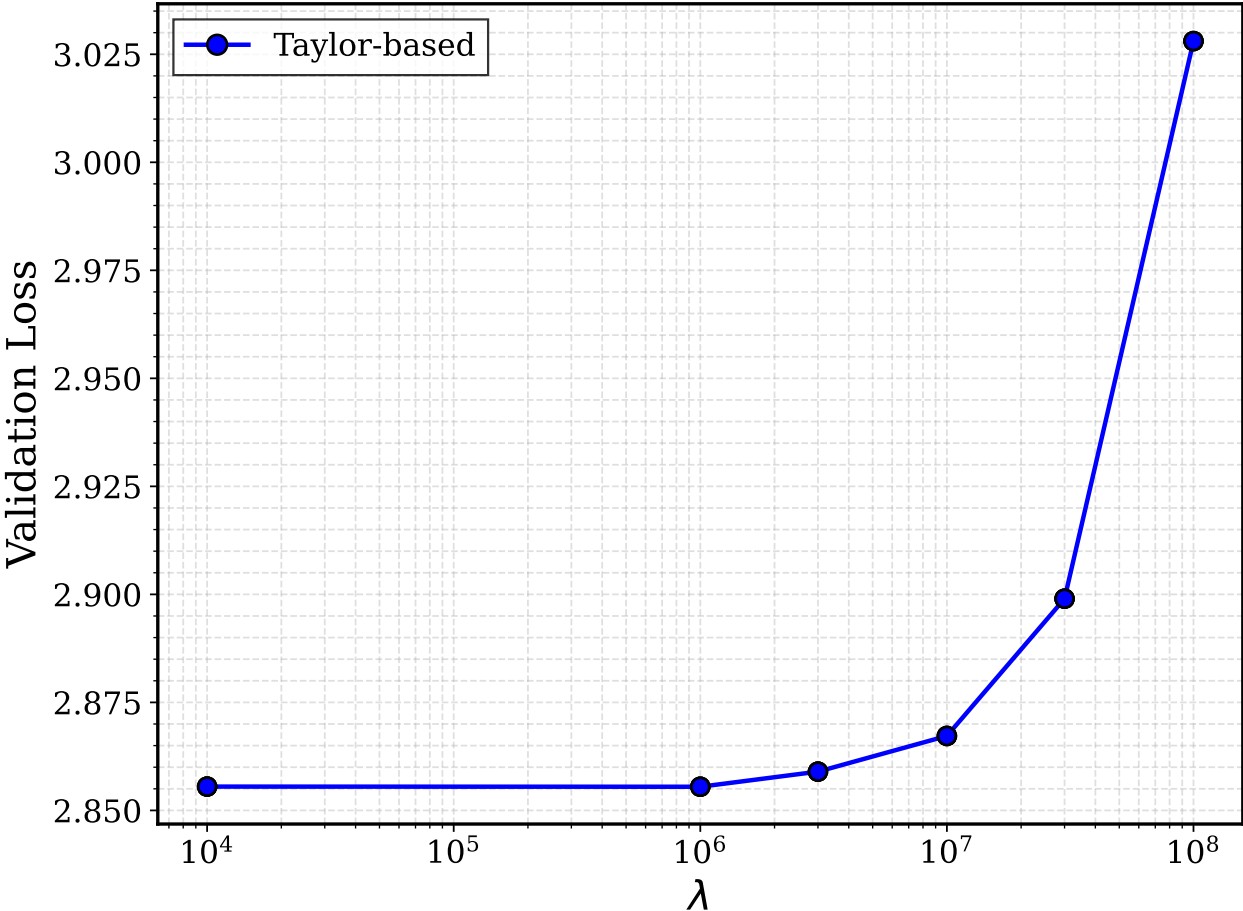

**Figure 8.** Final validation loss versus the Taylor correction coefficient $\lambda$ for DC-ASGD-style compensation on SmoLLM-135M with Muon. None of the tested coefficients improves over the standard delayed baseline.

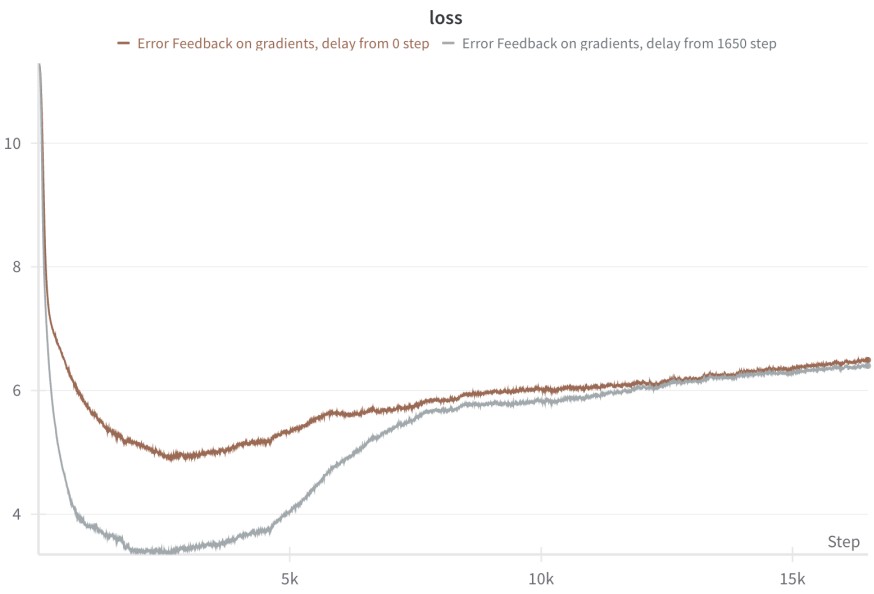

**Figure 9.** Gradient-level Error Feedback on the 135M model. Unlike the update-level correction used in the main experiments, applying the correction directly to raw gradients leads to divergence in our setting.

### D.6. Interaction of PipeDream Schedule with Error-Feedback

We provide results on the interaction between the PipeDream schedule (variable delay) and our Error-Feedback mechanism in Table 8. Consistent with our findings in Section 3.2, Error-Feedback yields consistent improvements for robust optimizers like Muon and SOAP. However, it leads to a slight degradation for Nadam.

**Table 8.** Comparison of synchronous training, PipeDream-2BW with constant one-step delay, and the original PipeDream schedule with variable delay on the 135M model. For each asynchronous schedule, we report results without and with Error Feedback. Values in parentheses denote the loss gap relative to the corresponding synchronous baseline. Bold entries indicate the smallest gap within each asynchronous column.

| Optimizer | Sync | 2BW | | PipeDream, $P=4$ | | PipeDream, $P=8$ | | PipeDream, $P=16$ | |
|---|---|---|---|---|---|---|---|---|---|
| | | w/o EF | w EF | w/o EF | w EF | w/o EF | w EF | w/o EF | w EF |
| Nadam | 2.891 | 2.936 (+0.045) | 2.955 (+0.064) | **2.889** (-0.002) | 2.905 (+0.014) | 2.910 (+0.019) | 2.918 (+0.027) | 2.950 (+0.059) | 2.940 (+0.049) |
| Muon, 0.95 | 2.839 | 2.856 (+0.017) | 2.845 (+0.006) | 2.861 (+0.022) | 2.858 (+0.019) | 2.881 (+0.042) | 2.877 (+0.038) | 2.917 (+0.078) | 2.914 (+0.075) |
| Muon, 0.99 | 2.841 | **2.855** (+0.014) | **2.842** (+0.001) | 2.844 (+0.003) | **2.840** (-0.001) | **2.857** (+0.016) | **2.854** (+0.013) | **2.882** (+0.041) | **2.876** (+0.035) |
| SOAP | 2.855 | 2.868 (+0.013) | 2.854 (-0.001) | 2.864 (+0.009) | 2.858 (+0.003) | 2.881 (+0.026) | 2.864 (+0.009) | 2.910 (+0.055) | 2.904 (+0.049) |

### D.7. Learning Rate Robustness for the 2B Model

Finally, to verify the stability of our method at the 2B scale, we present additional training runs varying the peak learning rate across training horizons in Figure 10.

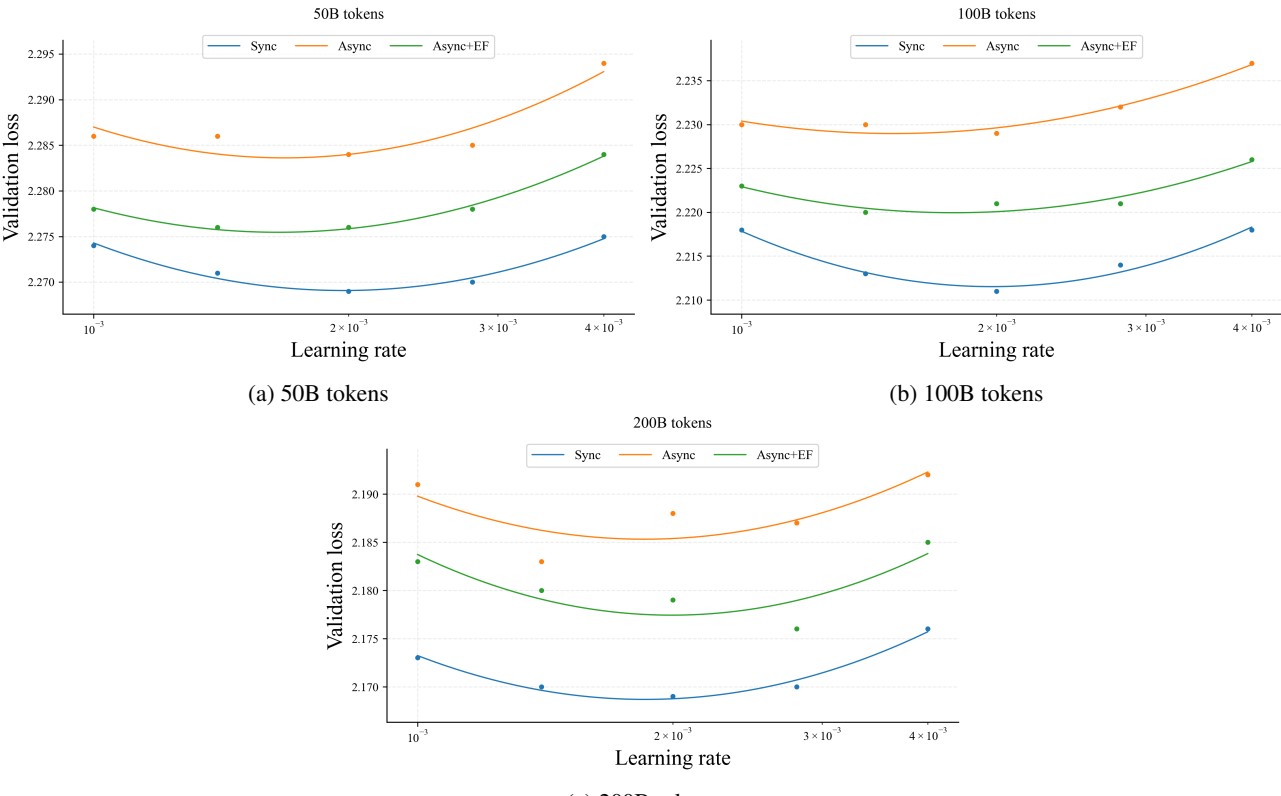

(a) 50B tokens

(b) 100B tokens

(c) 200B tokens

**Figure 10.** Loss as a function of learning rate for synchronous and delayed 2B MoE training at different scales.

### D.8. Effect of $\beta_2$ on the Synchronous-Start Loss Spike

We show in Figure 11 that the spike in train loss is notably larger for $\beta_2 = 0.999$ in comparison to $\beta_2 = 0.95$.

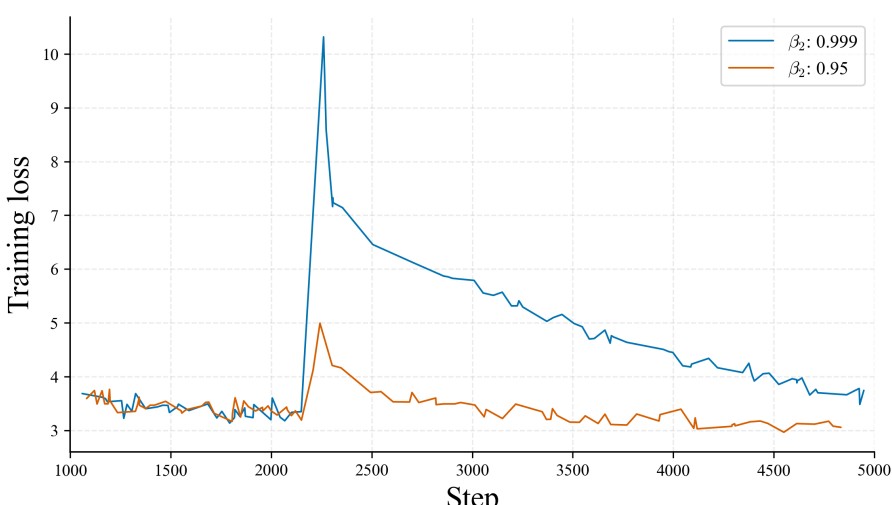

**Figure 11.** The relationship between the $\beta_2$ value and the loss "spike".

## E. Memory and Runtime Overhead

### E.1. Memory overhead

PipeDream-2BW and Error Feedback each introduce one additional parameter-sized state. For PipeDream-2BW, this state is the extra parameter version required to preserve forward–backward consistency under delayed updates. For Error Feedback, it is the residual buffer used to accumulate the correction. In modern large-scale training setups, however, this overhead is

applied to the local model shard stored on each GPU, not to the full model. As a result, the per-GPU cost is typically small because model states are distributed across pipeline, tensor, expert, and data-parallel dimensions.

We first consider DeepSeek-V3 (Liu et al., 2024b), a 681B-parameter MoE model trained on 2048 GPUs with 16-way Pipeline Parallelism (PP) and 64-way Expert Parallelism (EP). The remaining data-parallel degree is therefore $2048/(16 \cdot 64) = 2$. DeepSeek-V3 has 61 hidden layers in total, so each pipeline stage stores at most four layers. For a MoE layer, each GPU stores all non-expert components assigned to its pipeline stage, the shared expert, and only $256/64 = 4$ routed experts due to expert parallelism. This gives approximately $0.409$B parameters per MoE layer per GPU, or about $4 \cdot 0.409$B $\approx 1.6$B parameters per GPU for four MoE layers. Since DeepSeek-V3 uses ZeRO-1 with data-parallel degree 2, an additional FP32 master-weight copy is sharded across two data-parallel ranks, giving an estimated cost of

$$1.6\text{B} \cdot 4 \text{ bytes}/2 \approx 3.2 \text{ GB}$$

per GPU. On 80GB GPUs, this overhead is not prohibitive.

As a second example, consider LLaMA 3 405B (Grattafiori et al., 2024), which uses 8-way tensor parallelism and 16-way pipeline parallelism. Before FSDP sharding, the resident parameter count per GPU is approximately

$$405\text{B}/(8 \cdot 16) \approx 3.16\text{B}.$$

With an effective FSDP sharding factor of about $128$ for optimizer and master-weight states, one additional FP32 sharded state costs

$$3.16\text{B} \cdot 4 \text{ bytes}/128 \approx 0.10 \text{ GB}$$

per GPU. In this setting, the additional memory overhead is therefore negligible.

The same conclusion holds in our largest experiment: a 10B-parameter MoE model trained on 64 GPUs. Since our setup partitions all hidden layers across devices and does not replicate sublayers across GPUs, each GPU stores at most about 200M master-weight parameters, allowing a small margin for embeddings and the language-model head. Thus, one additional FP32 parameter-sized state costs approximately

$$200\text{M} \cdot 4 \text{ bytes} = 800 \text{ MB}.$$

In practice, the total additional memory cost of Async PP with Error Feedback was below 1.5GB per GPU, which is less than $2\%$ of an 80GB GPU. These estimates suggest that as long as the number of additional parameter-sized states remains a small constant and does not grow with pipeline depth, the memory overhead of PipeDream-2BW and Error Feedback is not a major obstacle in realistic LLM training scenarios.

### E.2. Runtime overhead

The main runtime advantage of Async PP comes from eliminating pipeline bubbles. We estimate this effect using the bubble model from the DeepSeek-V3 technical report (Liu et al., 2024b). Let $P$ denote the pipeline depth, $M$ the number of micro-batches, $F$ the forward time for one micro-batch chunk, $B$ the backward time, $W$ the backward-for-weights component, and $F\&B$ the execution time of an overlapped forward/backward pair. We define the bubble-to-compute ratio as

$$\rho = \frac{T_{\text{bubble}}}{T_{\text{compute}}}, \qquad T_{\text{compute}} = M(F + B).$$

Async PP has no pipeline bubbles in this schedule-level model, so $\rho_{\text{async}} = 0$. Thus, $1 + \rho$ can be interpreted as the slowdown of a synchronous PP schedule relative to the async ideal.

For standard synchronous schedules, the DeepSeek-V3 technical report gives the following bubble ratios:

$$\rho_{\text{1F1B}} = \frac{P - 1}{M},$$

$$\rho_{\text{ZB1P}} = \frac{(P - 1)(F + B - 2W)}{M(F + B)},$$

$$\rho_{\text{DualPipe}} = \frac{\left(\frac{P}{2} - 1\right)(F\&B + B - 3W)}{M(F + B)}.$$

To keep the analysis simple, we assume that communication is fully overlapped, so $F\&B \approx F + B$. We consider two standard zero-order compute models. The first assumes $B = 2F$ and $W = F$, corresponding to matmul-only accounting. The second assumes $B = 3F$ and $W = F$, roughly accounting for activation recomputation on the input-gradient path. We report results for $P = 16$, a pipeline depth used in modern large-scale training setups such as DeepSeek-V3 (Liu et al., 2024b) and LLaMA 3 (Grattafiori et al., 2024).

**Table 9.** Slowdown factors $1 + \rho$ relative to the async ideal under the DeepSeek-V3 bubble model with $P = 16$.

| Schedule | $M = 16$ | $M = 32$ | $M = 64$ |
|---|---|---|---|
| Async PP / PipeDream-2BW | 1.000 | 1.000 | 1.000 |
| 1F1B, $B = 2F$ | 1.938 | 1.469 | 1.234 |
| ZB1P, $B = 2F, W = F$ | 1.313 | 1.156 | 1.078 |
| DualPipe, $B = 2F, W = F$ | 1.292 | 1.146 | 1.073 |
| 1F1B, $B = 3F$ | 1.938 | 1.469 | 1.234 |
| ZB1P, $B = 3F, W = F$ | 1.469 | 1.234 | 1.117 |
| DualPipe, $B = 3F, W = F$ | 1.438 | 1.219 | 1.109 |

Under these standard bubble models, synchronous PP can still incur substantial schedule-level overhead at practical micro-batch counts, whereas Async PP removes this bubble term entirely. This analysis is not a substitute for end-to-end wall-clock measurements, which depend on implementation details, communication overlap, and hardware. Nevertheless, it provides a simple estimate of the runtime advantage that can be expected from eliminating synchronous pipeline bubbles.

