# OpenReview forum: "One-Step Gradient Delay is Not a Barrier for Large-Scale Asynchronous Pipeline Parallel LLM Pretraining"
_ICML.cc/2026/Conference — ICML 2026 regular_

### Official Review · Reviewer_AzXm · 2026-03-10

**Soundness:** 3
**Presentation:** 3
**Significance:** 3
**Originality:** 3
**Overall Recommendation:** 5
**Confidence:** 3

**Summary:**

This paper re-evaluates the viability of Asynchronous Pipeline Parallelism (Async PP) for large-scale LLM pretraining, challenging the assumption that optimization under gradient staleness is fundamentally unstable. The authors argue that the severe convergence degradation historically associated with Async PP is largely an artifact of relying on older optimizers like AdamW. By leveraging the PipeDream-2BW schedule to guarantee a constant one-step gradient delay, the study provides extensive empirical evidence showing that modern optimizers (like Muon) are inherently resilient to staleness. To further eliminate the performance gap, the submission proposes an optimizer-agnostic Error-Feedback mechanism that retrospectively corrects the optimizer's updates.

**Compliance With Llm Reviewing Policy:**

Affirmed.

**Final Justification:**

This paper re-evaluates Asynchronous Pipeline Parallelism by showing that historical convergence issues are largely artifacts of using older optimizers like AdamW rather than inherent limitations of gradient staleness. The benchmarking of modern algorithms is a major strength, particularly the identification of high momentum as a universal predictor of robustness. I find the proposed Error-Feedback mechanism both elegant and effective at bridging the final performance gap with synchronous training. Although I had initial concerns regarding wall-clock throughput and memory overhead, the authors provided a principled bubble model simulation and specific profiling that showed a small memory cost for their 10B model experiments. Since they achieved the successful demonstration of parity with synchronous training at this scale, I believe the work is technically solid and highly relevant for future large-scale pretraining. Therefore, I keep my positive score.

**Key Questions For Authors:**

Q1 (**actual speedup**): While the paper notes Async PP maximizes throughput by eliminating bubbles, the charts plot loss against optimization steps rather than time. Could the authors provide wall-clock time comparisons or Model FLOPs Utilization (MFU) metrics to quantify the actual real-world speedup?

Q2 (**memory overhead**): PipeDream-2BW already requires storing an extra copy of model parameters. Since the proposed Error-Feedback mechanism needs to store the previous update to compute the current one, what is the exact VRAM overhead of the Async+EF setup?

Q3 (**memory tradeoff**): Related to Q2, the Async-PP + EF method requires extra memory for parameter copies and update buffers. If a practitioner used that same memory to simply increase the batch size via ZeRO/FSDP (thereby reducing the total number of updates and amortizing bubble overhead), would Async-PP still provide a meaningful wall-clock speedup?

**Limitations:**

yes

**Strengths And Weaknesses:**

Strength:
1. Instead of just testing the default AdamW, the authors conduct a broad evaluation across a variety of modern optimizers (Muon, SOAP, Adan, MARS, etc.). This effectively isolates the optimizer itself as the root cause of historical instability.
2. The authors successfully identify high momentum as a universal predictor of robustness against gradient staleness. Furthermore, their proposed Error-Feedback (EF) mechanism is elegant, optimizer-agnostic, and practically effective at closing the performance gap.

Weakness:
1. The main idea of adopting Async PP is to eliminate GPU idle time and maximize throughput. However, the results focus almost exclusively on validation loss and training steps. The paper lacks concrete wall-clock time comparisons or FLOPs metrics to demonstrate the actual real-world speedup achieved over the synchronous baseline.
2. The method relies on the PipeDream-2BW schedule, which ensures a constant one-step delay but requires keeping an additional copy of the model parameters in memory. Also, the GPU memory used for supporting the error feedback mechanism is barely analyzed.

---

> ### Author Rebuttal · Authors · 2026-03-31
>
> We thank the reviewer for their feedback and are glad that the reviewer appreciated our extensive empirical analysis of the convergence dynamics of different optimizers under delay and their relationship with the momentum parameter. We address the reviewer's concerns and questions below.
>
> **W1-Q1.**
>
> While we agree that additional throughput measurements would further strengthen the paper, we would like to emphasize that this was not the main focus of our work. Our goal was not to propose a new scheduling method, but rather to study how much the training quality of the existing Async PP approach can be improved through the use of different optimizers in the delayed setting. We believe we were able to show that, by using Muon and Error Feedback, one can achieve the same quality in terms of the number of optimizer steps.
>
> More detailed throughput-oriented measurements were conducted in other papers, for example in [1, Figure 5], [2, Figure 6], and [3, Figure 6]. Note that the speedup depends strongly on the specific setup used by the practitioner; for example, it depends on the number of stages in the pipeline, as well as to the number of microbatches in the global batch between two consecutive parameter updates in synchronous PP.
>
> **W2-Q2.**
>
> Indeed, both PipeDream-2BW and Error Feedback introduce one additional copy of the parameters required for training. However, in most modern training pipelines this additional overhead is not critical, because model-state-related memory is distributed across all GPUs and therefore the per-GPU cost remains fairly small. As an example, consider two training setups: DeepSeek-V3 681B [4] and LLaMA 3 405B [5].
>
> DeepSeek-V3 is trained with 16-way Pipeline Parallelism (PP) and 64-way Expert Parallelism (EP) on 2048 GPUs, with the remaining DP degree equal to 2048 / (16 * 64) = 2.
> The model has 61 hidden layers in total, and because PP = 16, each stage stores at most 4 layers (for simplicity, we only consider the MoE layers below). Each MoE layer has 256 routed experts and 1 shared expert; because EP = 64, each GPU stores only 256 / 64 = 4 routed experts for each MoE layer residing on its PP rank.
> Thus, on one GPU, one MoE layer contributes: (1) all of its attention weights, (2) all of its gate weights, (3) all norm weights, (4) the 1 shared expert, and (5) only 4 routed experts (256 / EP). This amounts to about 0.409B parameters per MoE layer per GPU. In total, this gives approximately 4 * 0.409B ~= 1.6B parameters per GPU.
> Now, since DP = 2 and DeepSeek uses ZeRO-1, the master weights are sharded across 2 GPUs, so the final estimate for one additional parameter copy is just 1.6B * 4 bytes / 2 ~= 3.2 GiB per GPU.
> Given that 80GB GPUs were used for training, these additional 3GB do not appear to be a prohibitive obstacle.
>
> LLaMA 3 405B is trained with 8-way tensor parallelism (TP) and 16-way PP. Thus, before FSDP sharding, the resident parameter count per GPU is approximately 405B / (8 * 16) = 3.16B parameters. The effective FSDP sharding factor relevant for optimizer/master states is about 128. Hence, one additional FP32 sharded state copy costs 3.16B * 4 bytes / 128 = 0.01 GB per GPU.
> In the case of LLaMA 3, the additional overhead is therefore entirely negligible.
>
> Thus, as long as the additional memory does not exceed some small constant number of extra model copies and does not grow linearly with pipeline depth, it is not a major issue in modern training scenarios.
>
> We agree that the paper would benefit from a more detailed discussion of this point, and we will add it to the camera-ready version.
>
> **Q3.**
>
> The calculations for Q2 show that, by abandoning Async PP in favor of synchronized training, a practitioner would typically save only a few GB. Moreover, in standard large-scale training, the majority of memory (for DeepSeek-V3 and LLaMA 3, approximately more than 65GB out of 80GB) is already consumed by activations. At the same time, the typical micro-batch size (the number of sequences processed simultaneously by one GPU) is usually already on the order of just a few sequences, or even exactly 1 sequence per GPU. Therefore, the additional memory required for the activations of one more sequence is likely to exceed 5GB. As a result, the memory freed by removing the extra buffers required for Async PP cannot be straightforwardly reused simply by increasing the batch size.
>
> [1] Ajanthan et al., Nesterov Method for Asynchronous Pipeline Parallel Optimization, ICML 2025.
>
> [2] Guan et al., "PipeOptim: Ensuring effective 1F1B schedule with optimizer-dependent weight prediction." IEEE Transactions on Knowledge and Data Engineering, 2025.
>
> [3] Narayanan et al., Memory-Efficient Pipeline-Parallel DNN Training, ICML 2025.
>
> [4] Liu et al., DeepSeek-V3 Technical Report, arXiv preprint 2024.
>
> [5] Grattafiori et al., The Llama 3 Herd of Models, arXiv preprint, 2024.

---

> > ### Author Rebuttal · Reviewer_AzXm · 2026-04-03
> >
> > I thank the authors for their responses, and my concerns are partially resolved.
> >
> > I'm convinced by the response to W3-Q3. For W1-Q1 and W2-Q2, I still think it should be important to include the actual runtime and the memory usage (or at least a careful analysis or a profiling result) in the environment that the authors work on.  Since it is stated in the Introduction section that asynchronous PP should be a method to alleviate the "bubbles" and to improve the training efficiency, and the paper investigated these staleness mitigation strategies, the benefits/drawbacks in terms of runtime and memory constraints should also be included.
> >
> > Therefore, I will keep my score as it is.

---

> > > ### Author Response · Authors · 2026-04-05
> > >
> > > > I still think it should be important to include the actual runtime and the memory usage (or at least a careful analysis or a profiling result) in the environment that the authors work on.
> > >
> > > **Runtime**
> > >
> > > Unfortunately, we cannot report wall-clock speedups of async PP vs sync PP from our current experiments, because we did not implement synchronous PP baselines in our codebase. Our goal in this work was to study optimization under delay rather than pipeline-system engineering, so we used standard DDP/FSDP implementation for the synchronous baseline. We agree that real runtime numbers for a sync PP baseline are important, and we will add them in the camera-ready by implementing sync PP in the same environment.
> > >
> > > For now, however, we can provide a careful theoretical analysis using the bubble model from Table 2 of the DeepSeek-V3 technical report [1]. Let $PP$ be the pipeline depth, $M$ the number of microbatches, $F$ the forward time of one microbatch chunk, $B$ the backward time, $W$ the “backward-for-weights” part, and $F\\&B$ the execution time of an overlapped forward/backward pair. We define the bubble-to-compute ratio as
> > > $
> > > \rho = \frac{T_{\text{bubble}}}{T_{\text{compute}}},  T_{\text{compute}} = M(F + B).
> > > $
> > > Async PP has no bubbles by construction, so $\rho_{\text{async}} = 0$; thus, $1 + \rho$ can be interpreted as the slowdown of a synchronous PP schedule relative to the async ideal. DeepSeek-V3 gives
> > > $
> > > \rho_{1F1B} = \frac{PP - 1}{M},
> > > $
> > > $
> > > \rho_{ZB1P} = \frac{(PP - 1)(F + B - 2W)}{M(F + B)},
> > > $
> > > $
> > > \rho_{DualPipe} = \frac{(\frac{PP}{2} - 1)(F\\&B + B - 3W)}{M(F + B)}.
> > > $
> > >
> > > To keep the analysis simple, we consider the idealized case where communication is fully overlapped (hidden), i.e. $F\\&B \approx F + B$. This simplification is conservative for DualPipe and does not advantage PipeDream-2BW, since it also always has a forward/backward pattern and can exploit the same type of interleaving as DualPipe (see Figure 4 in [1]). We further consider two standard approximation models: (i) $B = 2F, W = F$, corresponding to the usual matmul-only accounting; and (ii) $B = 3F, W = F$, which roughly accounts for activation recomputation/checkpointing on the input-gradient path. We report $PP = 16$, which is representative of modern large-scale training; for example, both DeepSeek-V3 and Llama 3 [2] use $PP = 16$.
> > >
> > > Using the aforementioned approximations, we obtain the following slowdown ratios:
> > >
> > > | Schedule | $M=16$ | $M=32$ | $M=64$ |
> > > |---|---:|---:|---:|
> > > | Async (PipeDream-2BW) | 1.0 | 1.0 | 1.0 |
> > > | 1F1B-sync (assuming $B=2F$) | 1.938 | 1.469 | 1.234 |
> > > | ZB1P (assuming $B=2F$) | 1.313 | 1.156 | 1.078 |
> > > | DualPipe (assuming $B=2F$) | 1.292 | 1.146 | 1.073 |
> > > | 1F1B-sync (assuming $B=3F, W=F$) | 1.938 | 1.469 | 1.234 |
> > > | ZB1P (assuming $B=3F, W=F$) | 1.469 | 1.234 | 1.117 |
> > > | DualPipe (assuming $B=3F, W=F$) | 1.438 | 1.219 | 1.109 |
> > >
> > > Thus, under standard bubble models, synchronous PP can still incur substantial overhead at practical microbatch counts, whereas async PP removes this term entirely. This theoretical analysis is not a substitute for real end-to-end measurements, but it provides a principled estimate of the expected runtime advantage of async PP; we will also include wall-clock comparisons in the camera-ready version.
> > >
> > > **Memory**
> > >
> > > Regarding memory overhead in our experimental setup, for our largest model (10B parameters trained on 64 GPUs), the additional memory cost of AsyncPP + EF was below 1.5 GB per GPU - which is less than 2% for a 80GB GPU.
> > >
> > > This is consistent with the following estimate. Unlike DeepSeek, we do not replicate any sublayers across GPUs; all the hidden layers are equally partitioned across devices. As a result, for a 10B model trained on 64 GPUs, each GPU stores no more than approximately 200M master-weight parameters (allowing a small margin for embeddings and the LM head). Therefore, each additional copy of the master weights requires approximately $200M * 4\text{bytes} = 800\text{ MB}.$
> > >
> > > We agree that clarity would be improved by explicitly reporting these overheads, and we will add them in the revised version.
> > >
> > > [1] Liu et al., DeepSeek-V3 Technical Report, arXiv preprint 2024.
> > >
> > > [2] Grattafiori et al., The Llama 3 Herd of Models, arXiv preprint, 2024.

---

### Official Review · Reviewer_QuRu · 2026-03-12

**Soundness:** 2
**Presentation:** 1
**Significance:** 2
**Originality:** 2
**Overall Recommendation:** 4
**Confidence:** 4

**Summary:**

This paper studies the problem of improving 1 step gradient delay in the context of PipeDream-2BW. The authors show that Muon is better as well as a correction term they propose.

**Compliance With Llm Reviewing Policy:**

Affirmed.

**Final Justification:**

The authors rebuttal have largely addressed my concerns which were related to  practical demonstration of time benefits over existing well adopted methods which they have illustrated with a bubble model which I believe clarifies the benefits and would be useful to have in the paper. The results are at a sufficient scale and demonstrate closing a performance gap in a non-trivial setting in particular the 10B MOE with 200B tokens. The memory is a slight concerns, especially with activation checkpointing (not sure if this was covered in the memory simulation). On the other hand Muon has fewer accumulators to start with  and certainly if utilization can be increased significantly practitioners may be willing to tolerate the additional model copy overhead.

My other concerns were related to presentation and writing which has largely been addressed however it is challenging to fully judge without seeing a revision draft. I believe the revisions proposed in the rebuttal are not negligible but very doable as they do not require new experiments or new theory. It seems a general issue in this years review process is extensive rebuttal discussions have been encouraged without the corresponding possibility to post revision manuscript for reviewers to consider in context thus I think the AC should take that into consideration.

**Key Questions For Authors:**

See above

**Strengths And Weaknesses:**

Strengths

— The experimental results clearly suggest that a problem previously observed for AdamW, does not exist or is reduced when using Muon, which already has faster convergence. An important finding potentially for practical implementation of methods like PipeDream.

— The correction/EF term for 1-step delay appears to bring significant improvement while being novel

Weakness:

— There is no wall clock (even simulated) shown in the paper, and bandwidth constraints are not discussed. It is not clear if there would be significant improvement form doing this compared to 1F1B for example in modern HPC settings, however it is possible there can be benefits. This needs to be more clearly elucidated in the experiments and framing. Furthermore the memory overhead needs to be clear in these comparisons, assuming authors use weight stashing AND extra buffer with eF this seems a significant overhead.

— There are a lot of details in the practical implementation skipped, it seems that weight stashing would be necessary if the gradients are the true ones? What are the memory requirements in EF and non EF case?

— The technical novelty is somewhat limited in that the paper main focus is observing an interesting empirical property of Muon. However the EF approach to this appears to be novel however the discussion of this part is relatively limited. Indeed the EF seems to be a more significant technical contribution (it seems to work even with Adam) but it is presented as more auxiliary in the introduction/abstract.

— The “error feedback” name for the correction might not be the right term here as there is no ground truth error and may cause confusion with EF techniques in the literature, I would consider renaming this unless the connection to error feedback can be clarified.  In general given its importance in the results the presentation and motivation of the proposed correction can be improved.  Also there are a number of delay compensation approaches in the literature such as DC-ASGD that should be discussed and compared to the proposal. There is also weight prediction a detailed comparison with the proposal would be merited

— The theoretical contribution is limited and appears to only support that Muon can be compatible with existing theory on delayed gradient convergence but it does not seem to motivate the advantage of Muon compared to say Adam in delayed settings which is implied in the abstract.

---

> ### Author Rebuttal · Authors · 2026-03-31
>
> We thank the reviewer for their detailed comments. We appreciate their positive assessment of our Error Feedback (EF) method and experimental results, and respond to their questions below.
>
> Additional results that we will refer to with the apostrophe can be found at the anonymous link: https://anonymous.4open.science/r/async-pretrain-figures/figures.pdf.
>
> **W1-2:** Wall-clock time and memory overhead
>
> Due to the strict rebuttal space limit and the absence of a General Response option, we kindly refer the reviewer to (W1, Q1) and (W2, Q2) in our response to Reviewer AzXm, who raises essentially the same concern.
>
> **W2:** Implementation details
>
> Our work does not introduce new architectural components; we reuse the standard pipeline schedule from [1]. We agree that the paper would benefit from a more explicit description of PipeDream-2BW, and we will add such a discussion to the appendix. The only additions on our side are different optimizers and EF. The former is implemented straightforwardly, as in standard synchronous training, while the latter indeed requires storing the difference between consecutive updates. The training setup, optimizer hyperparameters, and model architectures are described in Appendix A.1 and A.2.
> If the reviewer has any additional specific questions about the setup, we would be happy to clarify them.
>
> **W3:** EF role
>
> We are grateful that the reviewer appreciated our EF method. At the same time, we would like to emphasize that our contribution is broader than EF alone. In addition to EF, we view the following as core contributions: (1) the observation that one-step-delayed optimization is stable for modern optimizers, especially Muon; (2) identifying the relationship between optimizer performance under staleness and the momentum decay ratio; and (3) extensive optimizer benchmarking under staleness across multiple scales, including 10B MoE models. We believe these contributions are also very important and therefore deserved dedicated space in the paper. However, the camera-ready page limit will be less restrictive, and we will place greater emphasis on the EF contribution.
>
> Regarding the name, we believe our formulation in Eq. 3 is well aligned with the Error Feedback framework and thus the name is justified. In particular, it is inspired by Eq. (20) in [2]. Define
> $v_t = C(u_t(g_t) + e_t) = u_{t-1}(g_{t-1}) + e_t$,
> where
> $e_t = (u_{t-1}(g_{t-1}) + e_{t-1}) - v_t = (u_{t-1}(g_{t-1}) + e_{t-1}) - Q(u_{t-1}(g_{t-1}) + e_{t-1}) = u_{t-1}(g_{t-1}) - u_{t-2}(g_{t-2})$,
> which leads to the proposed algorithm. We therefore believe that the EF interpretation is natural rather than purely terminological. We also welcome any specific suggestions from the reviewer regarding this interpretation.
>
> **W4:** Other delay compensation techniques
>
> Section 3 already compares our EF method against several relevant baselines, including synchronous warmup (“start=1W”) from [3] and DC-ASGD [4] (Lines 302-312). We also provide a more detailed DC-ASGD ablation in Figure 1 of the supplementary link above. As the results show, for small $\lambda$, DC-ASGD does not improve validation loss over the asynchronous baseline without delay correction, and for larger $\lambda$ it performs worse. We also report results for SAPipe, mentioned by reviewer idjV, in Table 1. To the best of our knowledge, these are the only baselines directly suitable for our setting. If the reviewer is aware of additional related methods, we would be very glad to learn about them and compare against them as well.
>
> **W5:** Theoretical contribution
>
> We acknowledge that the phrasing in the abstract may be confusing. By saying “We support this with theoretical analysis,” we mean that our work provides the first convergence guarantees for Linear Minimization Oracle (LMO) algorithms under gradient delay, both with and without EF. and that this theoretical result supports the empirical observation that "Muon exhibits inherent robustness" under delay.
>
> Our theory focuses on the LMO family rather than Adam or other adaptive methods. Adam appears difficult to analyze theoretically; for example, theory still only establishes convergence guarantees comparable to SGD. Moreover, AdamW is known to be unstable in some settings and may diverge [5], which we believe is reflected in our experiments.
>
> We also provide additional empirical results on the origin of the Adam–Muon performance gap in our response to reviewer 5ZTH.
>
> References:
>
> [1] Narayanan et al., Memory-Efficient Pipeline-Parallel DNN Training, ICML 2021.
>
> [2] Stich et al., The Error-Feedback Framework: Better Rates for SGD with Delayed Gradients and Compressed Updates, arXiv preprint, 2021.
>
> [3] Ren et al., ZeRO-Offload: Democratizing Billion-Scale Model Training, USENIX ATC 2021.
>
> [4] Zheng et al., Asynchronous Stochastic Gradient Descent with Delay Compensation, ICML 2017.
>
> [5] Reddi et al., On the Convergence of Adam and Beyond, arXiv preprint, 2019.

---

> > ### Author Rebuttal · Reviewer_QuRu · 2026-04-03
> >
> > - I did not see any wall clock times or bandwith in the link sent.
> > - PipeDreamBW is a critical base for this work, and to  understand the proposal it needs to be presented alongside this with the weight stashing and all, it should be in the algorithm block etc. The precise method is hard to follow due to this.
> > - Re: EF, understood, it might be worth to put the connection above in the appendix. The confusion is typically an accumulator over more than one step and you normally have the ground truth available earlier, here it arrives later.
> > - Re: Theory. I think the proposed phrasing is still problematic. "Muon exhibits inherent robustness" sounds like Muon > Adam/SGD on this. But the theory appears to show LMO has the similar theoretical properties for delay as  SGD, thus not explaining fully the key finding of the paper. This is fine as a limitation but it should be presented without an attempt to hint at more.
> >
> > I think the observation about Muon and EF method are novel and potentially useful but I believe the paper would benefit from a significant revision in the writing and presentation as well as wall clock times or at least simulations with bandwith constraints. Also the significant memory burden added by weight stashing and the additional accumulator is a concern and may nearly double the memory usage for muon? This can contradict some of the benefits of pipelining. A clearer view of tradeoffs are needed, the memory overhead should be more clearly presented and how it can be mitigated.

---

> > > ### Author Response · Authors · 2026-04-05
> > >
> > > **Wall-clock time measurements**
> > >
> > > Unfortunately, we cannot report wall-clock speedups of async PP vs sync PP from our current experiments, because we did not implement sync PP baselines in our codebase. Our goal in this work was to study optimization under delay rather than pipeline-system engineering, so we used standard DDP/FSDP for the sync baseline. We agree that real runtime numbers for a sync PP baseline are important, and we will add them in the camera-ready by implementing sync PP in the same environment.
> > >
> > > For now, however, we can provide a careful theoretical simulation using the bubble model from Table 2 of the DeepSeek-V3 [1]. Let $PP$ be the pipeline depth, $M$ the number of microbatches, $F$ and $B$ the forward time and backward time of one microbatch chunk, $W$ the “backward-for-weights” part, and $F\\&B$ the time of an overlapped forward/backward pair. We define the bubble-to-compute ratio as
> > > $
> > > \rho = \frac{T_{\text{bubble}}}{T_{\text{compute}}}, T_{\text{compute}} = M(F + B).
> > > $
> > > Async PP has no bubbles by construction, so $\rho_{\text{async}} = 0$; thus $1 + \rho$ can be interpreted as the slowdown of a sync PP schedule relative to the async ideal. DeepSeek-V3 gives
> > > $
> > > \rho_{1F1B} = \frac{PP - 1}{M},
> > > $
> > > $
> > > \rho_{ZB1P} = \frac{(PP - 1)(F + B - 2W)}{M(F + B)},
> > > $
> > > $
> > > \rho_{DualPipe} = \frac{(\frac{PP}{2} - 1)(F\\&B + B - 3W)}{M(F + B)}.
> > > $
> > >
> > > To keep the analysis simple, we consider the idealized case where communication is fully overlapped, i.e. $F\\&B \approx F + B$. We further consider two standard approximation models: (i) $B = 2F, W = F$ corresponding to matmul-only accounting; and (ii) $B = 3F, W = F$, which roughly accounts for recomputations caused be activation checkpointing. We report results for $PP = 16$, because both DeepSeek-V3 and Llama 3 [2] use it.
> > >
> > > Using the approximations above, we obtain the following slowdown ratios:
> > >
> > > | Model | $M=16$ | $M=32$ | $M=64$ |
> > > |---|---:|---:|---:|
> > > | Async (PipeDream-2BW) | 1.0 | 1.0 | 1.0 |
> > > | 1F1B-sync (assuming $B=2F$) | 1.938 | 1.469 | 1.234 |
> > > | ZB1P (assuming $B=2F$) | 1.313 | 1.156 | 1.078 |
> > > | DualPipe (assuming $B=2F$) | 1.292 | 1.146 | 1.073 |
> > > | 1F1B-sync (assuming $B=3F, W=F$) | 1.938 | 1.469 | 1.234 |
> > > | ZB1P (assuming $B=3F, W=F$) | 1.469 | 1.234 | 1.117 |
> > > | DualPipe (assuming $B=3F, W=F$) | 1.438 | 1.219 | 1.109 |
> > >
> > > Thus, under standard bubble models, sync PP can still incur substantial overhead at practical microbatch counts, whereas async PP removes this term entirely. This theoretical analysis is not a substitute for real end-to-end measurements, but it provides an estimate of the expected runtime advantage of async PP; we will also include wall-clock comparisons in the camera-ready.
> > >
> > > **Paper writing and presentation**
> > >
> > > We will incorporate the proposed changes, including a more detailed discussion of PipeDream-2BW in the main paper, and a more explicit discussion of EF. We believe that one additional page in the camera-ready will allow us to do this properly.
> > >
> > > **Theory and phrasing**
> > >
> > > We acknowledge that our theory does not show that Adam is poor under delay; rather, it only shows that Muon can tolerate delay, in the same sense as SGD. We will make this explicit and clarify that our theoretical contribution is to establish that Muon is robust to delay, and that there is no theoretical result for Adam (although our experiments suggest it may be less robust). We will also revise the abstract accordingly, e.g., "We provide theoretical analysis showing convergence for Muon with and without EF."
> > >
> > > **Memory overhead**
> > >
> > > We would also like to kindly reiterate that while weight stashing and EF indeed introduce an additional copy of the parameters, our analysis indicates that this overhead is small and unlikely to be a practical issue (see please discussion in W2-Q2 in our initial response to reviewer AzXm). An additional indirect argument is that DeepSeek also explicitly used an extra copy of the parameters in DualPipe to enable bidirectional pipeline scheduling necessary to reduce bubble (see Section 3.2.1 and Table 2 in [1]).
> > >
> > > Also, in our experiments on a 10B model trained on 64 GPUs, the additional memory overhead of async PP+EF was at most 1.5GB (<2% for a 80GB GPU). For a 10B model trained on 64 GPUs, each GPU stores no more than approximately 200M master-weight parameters (accounting for a small margin for embeddings and the LM head), so each additional copy requires approximately
> > > $
> > > 200\text{M} * 4\text{ bytes} = 800\text{ MB}.
> > > $
> > >
> > > Therefore, we respectfully disagree that this constitutes a "significant memory burden." However, we will add a more detailed discussion of this trade-off in the camera-ready.
> > >
> > > ---
> > >
> > > We believe that we addressed all the reviewer's concerns and questions, none of which is a serious issue with our approach. Therefore, we would kindly request the reviewer to reconsider their score.
> > >
> > > [1] Liu et al., DeepSeek-V3 Technical Report, arXiv preprint, 2024.
> > >
> > > [2] Grattafiori et al., The Llama 3 Herd of Models, arXiv preprint, 2024.

---

### Official Review · Reviewer_idjV · 2026-03-13

**Soundness:** 3
**Presentation:** 3
**Significance:** 2
**Originality:** 2
**Overall Recommendation:** 5
**Confidence:** 3

**Summary:**

This paper proposes using Error-Feedback mechanism to mitigate delay effects in asynchronous Pipeline Parallelism. Theoretical analysis of convergence is provided given the assumption of smoothness and star convexity. Experiments show that the proposed algorithm bridges the performance gap with synchronous training.

**Compliance With Llm Reviewing Policy:**

Affirmed.

**Final Justification:**

The author's response resolves my concerns.
Although SAPipe achieves the results very similar to the the algorithm proposed in this paper, I think this paper has the novelty of using EF in PP.
In overall I think this is a good paper. Thus I increased the score.

**Key Questions For Authors:**

1. Is there any way to remove the assumption of star convexity?
2. For the other explored approaches mentioned in Section 3.3, such as delay compensation, are there any experiment results?

**Limitations:**

yes

**Strengths And Weaknesses:**

Strengths:
1. This paper proposes using Error-Feedback mechanism to mitigate delay effects in asynchronous Pipeline Parallelism.
2. Theoretical analysis of convergence is provided given the assumption of smoothness and star convexity.
3. Experiments show that the proposed algorithm bridges the performance gap with synchronous training.

Weaknesses:
1. The Assumption 2.1, smoothness + star convexity, is a very strong assumption.
2. For the baselines in the experiments, there are other variants of delay compensation that could be taken into account. For example: Chen, Yangrui, et al. "Sapipe: Staleness-aware pipeline for data parallel dnn training." Advances in neural information processing systems 35 (2022): 17981-17993.
3. Some important citations are missing, for example PipeSGD is exactly SGD with 1-step delay: Li, Youjie, et al. "Pipe-SGD: A decentralized pipelined SGD framework for distributed deep net training." Advances in Neural Information Processing Systems 31 (2018).

---

> ### Author Rebuttal · Authors · 2026-03-31
>
> We thank the reviewer for taking the time to review our work and we are glad that the reviewer appreciated our Error-Feedback delay mitigation strategy. We carefully address each of the reviewer's concerns below. Additional results that we will refer to with the apostrophe can be found at the anonymous link: https://anonymous.4open.science/r/async-pretrain-figures/figures.pdf.
>
> **W1, Q1:** Theoretical assumptions
>
> We acknowledge their concern regarding the strength of the theoretical assumptions used in the main text. However, we already provide a fully general convergence analysis for arbitrary-delay LMO algorithms under only smoothness, unbiasedness, and bounded variance of the gradient estimator (without convexity assumptions) in Appendix B (Theorem B.4). The star-convexity assumption appears only in the weight decay case (Theorem 2.2 / Theorem B.6) in order to obtain the clean exponential decay term, and it is a standard assumption in the literature [1], [2]. Moreover, [3] argues that star-convexity is a more natural property of neural network optimization landscapes than plain non-convexity.
>
> **W2, Q2:** Other delay mitigation techniques
>
> We thank the reviewer very much for pointing out the “SAPipe: Staleness-Aware Pipeline for Data-Parallel DNN Training” paper. It is indeed a highly relevant paper that we unfortunately overlooked, although it was originally designed for a somewhat different setting. We conducted experiments with this method on the 135M-parameter model for Muon, AdamW, and SOAP, and present the results in Table 1'. As can be seen, SAPipe indeed achieves results very close to those of our Error Feedback method.
>
> **W3:** Other one-step-delayed scenario
>
> We thank the reviewer again for the reference to Youjie Li et al., “Pipe-SGD: A Decentralized Pipelined SGD Framework for Distributed Deep Net Training,” which studies exactly the one-step-delay case for SGD. However, we note that the analysis in this paper is carried out in a distributed learning setting, while still being relevant to our work. We will cite this paper in the camera-ready version.
>
> [1] Understanding Gradient Orthogonalization for Deep Learning via Non-Euclidean Trust-Region Optimization, Kovalev, 2025
>
> [2] Stochastic Distributed Learning with Gradient Quantization and Double-Variance Reduction, Horvath et al., 2019
>
> [3] Methods for Convex (L0, L1)-Smooth Optimization: Clipping, Acceleration, and Adaptivity, Gorbunov et al., ICLR 2025

---

> > ### Author Rebuttal · Reviewer_idjV · 2026-04-01
> >
> > The author's response resolves my concerns.
> > Although SAPipe achieves the results very similar to the the algorithm proposed in this paper, I think this paper has the novelty of using EF in PP.
> > In overall I think this is a good paper. Thus I increased the score.

---

### Official Review · Reviewer_5ZTH · 2026-03-13

**Soundness:** 2
**Presentation:** 2
**Significance:** 3
**Originality:** 3
**Overall Recommendation:** 4
**Confidence:** 5

**Summary:**

The paper investigates asynchronous pipeline parallel training, specifically for PipeDream-2BW which only has a one-step delay. It shows that the delay does not cause severe degradation for all optimizers, only for some of them, and also proposes and investigates other strategies for mitigating the degradation.

**Compliance With Llm Reviewing Policy:**

Affirmed.

**Final Justification:**

I am glad the authors properly investigated why AdamW fails, which was a key concern of mine, their new investigation properly showed this key part that was missing in the original submission.

Like Reviewer QuRu I think that there was a lot of presentation issues (especially a lot of issues with the mathematics that I had to point out) that the authors have now addressed but need to incorporate. I also share QuRu's complaint: since the authors can't show an integrated version, the best we can do is hope that the authors properly integrate and proofread their final version.

As a result I am increasing my score to be positive about the paper. However, I am still a bit concerned by the repeated need to ask the authors to fix/improve their proof, which is why I am only advocating for an increase to weak accept.

**Key Questions For Authors:**

Have a few questions in the strengths and weaknesses section.

In PipeDream-2BW two copies of the weights are kept, is that the same in this paper?

**Limitations:**

Some limitations are discussed in the paper. However other limitations, like the fact that EF means storing an extra set of gradients (so basically another copy of the model weights), are not mentioned.

**Strengths And Weaknesses:**

It is very interesting that AdamW and MARS do so badly when there is a delay but the other optimizers still do quite well, this is an important observation. However, there is no investigation into why, which is very important. Usually for AdamW the issue is attributed to the second order ema (variance) being outdate and thus scaling the variance badly, however other optimizers that use this do ok, and MARS doesn't fit with this reason. It would be good to at least show some plots of gradient behaviour between the methods to see if there is any trends that explain why this phenomenon occurs.

Figure 4 shows the performance gap against momentum decay ratio for methods other than AdamW and MARS, any reason why not to include them? Perhaps the issue with those optimizers are the standard hyperparameters, there should have been hyperparameter investigations done on them as well. At the very least, experiments with AdamW and MARS with higher momentum should have been reported.

I like the idea of using error feedback for delay correction, though memory-wise it is costly. As mentioned, the EF can be calculated on the raw gradients or on the updates, there should be ablations showing that the latter is better, as theoretically it is not clear what the right approach is. Does the memory cost make training more difficult (slower, harder to fit on GPUs, etc.)?

I like that other explored approaches are mentioned and discussed.

In most of the experiments it is not clear what the number of stages is, which is a very important value for the main concern of this paper.

Minor weakness: I was surprised how little explanation was given to why staleness occurs, and why it is variable in the original pipedream and constant in the second version. This is a pretty important point, and is somewhat subtle and difficult to understand, I expected at least pointing towards a good reference in a previous paper or a section in the appendix with an explanation with equations or a diagram. The explanation in the paper starts off with expecting this is understood and diving into a formalization of delayed gradient update in Algorithm 1, while I was expecting a formal explanation of how that delayed update arises. Also not sure what the need was for Algorithm 2, isn't it clear from algorithm 1?

In table 2, it is not clear/explained what "start=1W" means.

In the proof for Lemma C.2, how do you get the line at the bottom of page 16?.

---

> ### Author Rebuttal · Authors · 2026-03-31
>
> We thank the reviewer for their detailed and thoughtful feedback. We are glad that they appreciated our experimental results, as well as our Error Feedback approach. We address their concerns and questions below.
>
> W1: Why does AdamW fail, while some other optimizers do not?
>
> Thank you for raising this very important question. To investigate the reasons for the observed behavior and better understand the origins of the performance gap between the optimizers, we conducted several additional experiments. Below, we present results from these targeted experiments aimed at studying AdamW instability in the delayed setting compared to Muon.
>
> First, we measured several metrics of discrepancy, such as cosine similarity and relative difference norms $(||x - y|| / ||x||)$, between stale updates and the corresponding correct updates (i.e., the updates that would have been applied if the non-delayed gradient had been used at that step) and presented the results in Figures 2'–3'. Somewhat surprisingly, these results show the opposite of what one might expect: both cosine similarity and relative difference norms are better for AdamW than for Muon.
>
> We also hypothesized that, similarly to [1], the issue might originate from the final LM head layer. However, even when we switched this final layer to synchronous mode, the result remained the same: without any additional error-correction mechanism, AdamW still performs much worse and reaches a loss above 3.0 on the 135M model.
>
> We also ran experiments in which the delay affects only the update of $m$, while the update of $v$ remains synchronous. Surprisingly, this did not improve the results at all, and the loss values were almost completely identical, which once again supports our finding in the paper that $\beta_1$/momentum is the decisive factor for optimization performance under staleness.
>
> In general, based on our results, we observe that each method has certain ranges of convenient $\beta_1$ values for operating under delay. For some methods, these favorable $\beta_1$ ranges begin at relatively large values, for example SOAP with $\beta_2 = 0.999$ (Figure 4), and AdamW appears to belong to this group. On the other hand, each method also has another safe region corresponding to stable training even without any staleness; and in the case of AdamW, it seems that the $\beta_1$ value required for delay robustness becomes too large for the method even in the synchronous setting. Experiments with synchronous AdamW at $\beta_1 = 0.99$ show substantial degradation compared to $\beta_1 = 0.9$ — 2.939 vs 2.877.
>
> We will add all these results to the camera ready version of the paper.
>
> W2: Momentum Decay Ratio plots (Fig. 4)
>
> In the paper, we do provide such plots for the set of optimizers excluding AdamW and MARS; the only reason was to preserve readability. The performance gap for these two methods (without Error Feedback or other delay-correction techniques) is substantially larger than for the remaining optimizers. Nevertheless, we observe the same qualitative trend with respect to the momentum coefficient ($\beta_1$ for AdamW/MARS).
>
> W3: EF strategies ablation
>
> We did indeed try applying EF to gradients; however, this approach simply diverges. An example loss curve can be found in Figure 4'.
>
> W3-Q1: Additional Memory Consumption for the Error Feedback mechanism
>
> Because of the official space constraints for each rebuttal and the absence of a General Response option, we would kindly ask you to refer to (W1, Q1) in our response to Reviewer AzXm, as they raise a very similar question.
>
> W4: Number of Stages
>
> As noted in Line 73, in the case of PipeDream-2BW the delay is always the same regardless of the number of stages, so mathematically the training dynamics should be identical for any number of stages. For PipeDream, where the number of stages does indeed affect the training dynamics, we report the number of stages used in Table 4.
>
> W5: Why staleness occurs
>
> We agree that the paper would benefit from a more detailed discussion of how asynchronous PP schemes operate, and we will add a more thorough description in the camera-ready version.
>
> W6: Algorithm 2 and Algorithm 1
>
> Algorithm 2 is presented purely for readability and convenience, as the specific Muon instantiation of the more general Linear Minimization Oracle (LMO) formulation given in Algorithm 3.
>
> W7: Proof Transition
>
> This transition indeed requires additional explanation for smooth reading. We apply the parallelogram inequality and then explicitly expand the squared norm for the first sum. Because the stochastic gradients $\xi_i$ are i.i.d. and the estimator is unbiased, all cross-product terms vanish in expectation, yielding the expression in Lines 874–878. We also thank you for noticing the typo (equality instead of inequality); this will be corrected in the camera-ready version.
>
> [1] Deconstructing What Makes a Good Optimizer for Autoregressive Language Models, Rosie Zhao, ICLR 2025

---

> > ### Author Rebuttal · Reviewer_5ZTH · 2026-04-04
> >
> > I have a few follow up questions/comments:
> >
> > - I was confused for a long time about the results the authors were talking about until I noticed the have provided a link for responses to other reviews.
> >
> > W1
> > - The results and discussion "based on our results, we observe that each method has certain ranges of convenient $\beta_1$ values for operating under delay" and "On the other hand, each method also has another safe region corresponding to stable training even without any staleness" is a good finding that was missing in the original version of the paper. Please make sure to include this with the full experimental investigation in your revised version.
> >
> > W3-Q1
> > - I think you mean W2, Q2 not W1, Q1?
> > - I appreciate the explanation, but think that it is important to report actual memory and time overhead results for when EF is used in your experimental setup.
> >
> > W7
> > - "Because the stochastic gradients $\xi_i$ are i.i.d. and the estimator is unbiased, all cross-product terms vanish in expectation" I'm not following why they are independent when taking into account the delay: $x_i$ has a dependency on $g(x_{prev(i)},\xi_{prev(i)})$ ? I think this needs to be treated a bit more carefully.

---

> > > ### Author Response · Authors · 2026-04-05
> > >
> > > Thank you for this comment, and we sincerely apologize for the confusion. We accidentally forgot to include the link to the results in our earlier response.
> > >
> > > **W1**
> > >
> > > We will definitely include all newly added experiments in the camera-ready version. We will also extend the analysis further to provide a more detailed dissection of AdamW dynamics under gradient delay.
> > >
> > > **W3-Q1**
> > >
> > > > I think you mean W2, Q2 not W1, Q1?
> > >
> > > Yes, absolutely - sorry for this typo.
> > >
> > > > it is important to report actual memory and time overhead results for when EF is used in your experimental setup
> > >
> > > From the runtime perspective, the overhead of EF is negligible, since applying EF only requires a small number of additional elementwise operations. This cost is dominated by the matrix multiplications in the forward and backward passes. For the same reason, the runtime cost of standard elementwise optimizer updates (e.g., AdamW-style updates) is typically negligible compared to the cost of computing the gradients themselves.
> > >
> > > Regarding memory overhead in our experimental setup, for our largest model (10B parameters trained on 64 GPUs), the additional memory cost of AsyncPP + EF was below 1.5 GB per GPU - which is less than 2% for a 80GB GPU.
> > >
> > > This is consistent with the following estimate. Unlike DeepSeek v3 [1] (see also our discussion in the initial response to reviewer AzXm), we do not replicate any sublayers across GPUs,  thus all the hidden layers are equally partitioned across devices. As a result, for a 10B model trained on 64 GPUs, each GPU stores no more than approximately 200M master-weight parameters (allowing a small margin for embeddings and the LM head). Therefore, each additional copy of the master weights requires approximately $200M * 4\text{bytes} = 800\text{ MB}.$
> > >
> > > We agree that clarity would be improved by explicitly reporting these overheads, and we will add them in the camera-ready version.
> > >
> > > **W7.**
> > >
> > > The reviewer is correct that $x_i$ (and therefore $\nabla f(x_i)$) depends on previous stochastic samples, so the relevant terms are not independent in a naive sense. Our argument does **not** rely on the mixed terms between the *stochastic-noise and drift parts* vanishing in expectation.
> > >
> > > Instead, we first decompose
> > >
> > > $$
> > > S_1 := \\sum_i \\alpha (1-\\alpha)^{k-i} \\bigl( g(x_{\\mathrm{prev}(i)}; \\xi_{\\mathrm{prev}(i)}) - \\nabla f(x_{\\mathrm{prev}(i)}) \\bigr)
> > > $$
> > >
> > > and
> > >
> > > $$
> > > S_2 := \\sum_i \\alpha (1-\\alpha)^{k-i} \\bigl( \\nabla f(x_{\\mathrm{prev}(i)}) - \\nabla f(x_i) \\bigr).
> > > $$
> > >
> > > We then use
> > >
> > > $$
> > > \\lVert S_1 + S_2 \\rVert_*^2 \\le 2 \\lVert S_1 \\rVert_*^2 + 2 \\lVert S_2 \\rVert_*^2.
> > > $$
> > >
> > > The cancellation of cross-terms is used only for $S_1$. More precisely, by Assumption B.3 we first pass to the Euclidean norm:
> > >
> > > $$
> > > \\lVert S_1 \\rVert_*^2 \\le \\rho^2 \\lVert S_1 \\rVert_2^2.
> > > $$
> > >
> > > Then we expand the Euclidean square. Define
> > >
> > > $$
> > > \\delta_j := g(x_j; \\xi_j) - \\nabla f(x_j)
> > > $$
> > >
> > > and
> > >
> > > $$
> > > \\mathcal F_{j-1} := \\sigma(\\xi_0, \\dots, \\xi_{j-1}).
> > > $$
> > >
> > > Then for $j > l$,
> > >
> > > $$
> > > \\mathbb E \\langle \\delta_j, \\delta_l \\rangle = \\mathbb E \\big[ \\mathbb E( \\langle \\delta_j, \\delta_l \\rangle \\mid \\mathcal F_{j-1} ) \\big].
> > > $$
> > >
> > > Since $\\delta_l$ is measurable with respect to $\\mathcal F_{j-1}$, and Assumption B.1 gives
> > >
> > > $$
> > > \\mathbb E[\\delta_j \\mid \\mathcal F_{j-1}] = 0,
> > > $$
> > >
> > > it follows that
> > >
> > > $$
> > > \\mathbb E \\big[ \\langle \\delta_j, \\delta_l \\rangle \\mid \\mathcal F_{j-1} \\big] = \\langle \\mathbb E \\big[ \\delta_j \\mid \\mathcal F_{j-1} \\big], \\delta_l \\rangle = 0.
> > > $$
> > >
> > > Thus, the cross-terms vanish in expectation for the stochastic part $S_1$. The same argument also covers the initial iterations where one may have $x_j = x_l$: the key point is not whether the iterates coincide, but that the earlier term depends only on the past and therefore does not depend on $\\xi_j$.
> > >
> > > In contrast, no analogous cancellation is used for the $S_2$ term. That part is bounded separately using Assumption B.2 together with the delay bound on the distance between $x_{prev}$ and $x_i$.
> > >
> > > To make this clearer, in the revised version we will explicitly state the order of the argument: first separate the stochastic and drift terms, then apply the norm-equivalence bound to $S_1$, and only afterwards expand the Euclidean square and use conditional expectation to remove the cross-terms.
> > >
> > > ---
> > >
> > > To sum up, we believe that we addressed all the reviewer's concerns and questions. Therefore, we would kindly ask the reviewer to reconsider their score.
> > >
> > > [1] Liu et al., DeepSeek-V3 Technical Report, arXiv preprint 2024.

---

### Decision · Program_Chairs · 2026-04-30

**Decision:**

Accept (regular)

**Comment:**

Based on the reviews, rebuttal, and discussion, I recommend acceptance. The paper makes a useful practical contribution by showing that the instability historically associated with asynchronous pipeline parallelism is not inherent to one-step delay, but depends strongly on optimizer choice, and by introducing an effective error-feedback mechanism that further reduces the gap to synchronous training.

At the same time, the final version should address the remaining concerns raised during discussion:

* Some proof details required clarification (especially a key inequality needs some corrections). This should be carefully checked and integrated in the final version.
* The paper should better explain the optimizer-specific behavior under delay, especially why AdamW degrades while other methods appear more robust.

Overall, this is a solid accept, with some remaining issues in presentation and analysis that should be addressed in the camera-ready.